# MYH10 activation rescues contractile defects in arrhythmogenic cardiomyopathy (ACM)

Nieves García-Quintáns[1], Silvia Sacristán[1], Cristina Márquez-López[1], Cristina Sánchez-Ramos [1], Fernando Martinez-de-Benito [1,2], David Siniscalco[3], Andrés González-Guerra [1], Emilio Camafeita [1,2], Marta Roche-Molina[1], Mariya Lytvyn[1], David Morera[1], María I. Guillen[1], María A. Sanguino [1], David Sanz-Rosa [1,2,4], Daniel Martín-Pérez[1], Ricardo Garcia [3] & Juan A. Bernal [1,2] ✉

The most prevalent genetic form of inherited arrhythmogenic cardiomyopathy (ACM) is caused by mutations in desmosomal *plakophilin-2* (*PKP2*). By studying pathogenic deletion mutations in the desmosomal protein PKP2, here we identify a general mechanism by which PKP2 delocalization restricts actomyosin network organization and cardiac sarcomeric contraction in this untreatable disease. Computational modeling of PKP2 variants reveals that the carboxy-terminal (CT) domain is required for N-terminal domain stabilization, which determines PKP2 cortical localization and function. In mutant PKP2 cells the expression of the interacting protein MYH10 rescues actomyosin disorganization. Conversely, dominant-negative MYH10 mutant expression mimics the pathogenic CT–deletion *PKP2* mutant causing actin network abnormalities and right ventricle systolic dysfunction. A chemical activator of non-muscle myosins, 4-hydroxyacetophenone (4-HAP), also restores normal contractility. Our findings demonstrate that activation of MYH10 corrects the deleterious effect of PKP2 mutant over systolic cardiac contraction, with potential implications for ACM therapy.

The most frequent genetic alterations in arrhythmogenic cardiomyopathy (ACM) affect the desmosomal gene *PKP2*, accounting for 40–60% of genotype-positive patients[1–3]. ACM is a genetic disease of the heart muscle[4,5] that predisposes to sudden cardiac death (SCD)[6], particularly in young patients and athletes[7,8]. To date all therapeutic interventions offered to these patients are palliative and not curative. The potential therapy for ACM will require new knowledge about the molecular mechanisms that drive the pathological development of the disease.

ACM, for many years known as arrhythmogenic right ventricular cardiomyopathy (ARVC)[9], shows autosomal-dominant inheritance of desmosomal proteins and usually manifests as impaired function of the right ventricle (RV)[10]. Desmosomes are dynamic intercellular junctions that maintain the structural integrity of skin and heart tissues by withstanding shear forces, linking the cellular edge to the internal cytoskeleton[11,12]. Actin is one of the major cytoskeletal proteins in eukaryotic cells and plays an essential role in several cellular processes, including mechano-resistance and contractile force generation[13]. Defective regulation of the organization of actin filaments in sarcomeres, owing to genetic mutations or deregulated expression of cytoskeletal proteins, is a hallmark of many heart and skeletal muscle disorders[14].

The ClinVar database (https://www.ncbi.nlm.nih.gov/clinvar/), part of the NCBI Entrez system, attempts to establish relationships between gene variants and phenotype. Remarkably, all nonsense *PKP2* variants are classified as pathogenic or likely pathogenic. Two

[1]Centro Nacional de Investigaciones Cardiovasculares (CNIC), Madrid, Spain. [2]CIBER de Enfermedades Cardiovasculares (CIBERCV), Madrid, Spain. [3]Materials Science Factory, Instituto de Ciencia de Materiales de Madrid (ICMM), CSIC, Madrid, Spain. [4]Universidad Europea, Madrid, Spain. ✉e-mail: jabernal@cnic.es

mechanisms can drive pathogenesis, on one hand, mRNAs containing premature stop codons may be eliminated by a surveillance pathway called nonsense-mediated mRNA decay that results in the reduction of mRNA levels and the haploinsufficient phenotype. Instead, translation of these aberrant mRNAs may lead to deleterious gain-of-function (GOF) or dominant negative activity of the resulting proteins. We previously demonstrated that PKP2 (c.2203 C > T), encoding the p.R735* mutant, which creates a premature stop codon and a C-terminal truncated PKP2 protein and is found in ACM patients from different families[4,15], can operate as a GOF disease mutant[16], underlining the functional importance of the PKP2 C-terminal domain. In general, pathogenic mutations in *PKP2* have usually been associated with the classical form of the disease that predominantly includes anomalous electrocardiograms, with structural abnormalities that lead to a progressive global RV contractile dysfunction[1].

Here we identified MYH10 actomyosin component as key differential interactor of PKP2 cardiac isoform using biochemical and mass spectrometry approaches. The main role of MYH10 protein is to orchestrate the mechanoenzymatic properties of stress fibers. Myosins execute numerous mechanical tasks in cells, including spatiotemporal organization of the actin cytoskeleton, adhesion, migration, cytokinesis, tissue remodeling, and membrane trafficking[17–21]. Using in vitro and in vivo models, we restore actin network and cardiac contraction by stimulating MYH10 activity in an ACM mouse model of disease. We show that localization of PKP2 C-terminal deletion mutants found in ACM patients from different families[4,15] alters actomyosin organization and function. We discovered that PKP2 mutants´ expression in cardiac cells led to functional alterations in MYH10 what is key to developing an ACM contractile phenotype. In these PKP2 mutants a rearrangement of the F-actin cytoskeleton paired with sarcomere dysfunction. Our findings illustrate how expression of mutant MYH10 disrupts actin network distribution and compromises RV cardiac systolic contraction in mice following a similar mechanism than p.R735* mutant when inducing ACM. Finally, we demonstrate that activation of non-muscle myosins with 4-hydroxyacetophenone (4-HAP) attenuates contraction disfunction observed in ACM animals.

## Results

### PKP2 interacts with actomyosin proteins

We have previously shown that C-terminal *PKP2* deletion mutant (c.2203C > T), encoding the PKP2-p.R735* protein operates as a GOF protein in arrhythmogenic cardiomyopathy (ACM) compromising RV function[16], although functional mechanistic details remain unresolved. To gain insight into the molecular basis that induce the contractile defects we generated and compared wild-type and mutant PKP2 protein interactomes. We initially identified cardiac specific PKP2a[22] binding partners by pull-down of Halo-tagged PKP2 proteins followed by mass spectrometry (MS) analysis. PKP2-HaloTag pull-down products included previously identified interacting partners such as the desmosomal components, plakoglobin (JUP), desmoplakin (DSP) and desmoglein (DGS) together with unknow interacting partners like cytoskeletal actin proteins (ACTA2 and ACTB), or myosin 9 (MYH9) and myosin 10 (MYH10), also known as non-muscle myosins NMIIA and NMIIB (Fig. 1a). To validate the results obtained using HaloTag-PKP2 variants we also performed an orthogonal immunoprecipitation (EGFP-tag) assay followed by MS analysis. In Fig. 1, we show a list of proteins identified by both independent techniques, confirming desmosomal and cytoskeletal proteins as strong PKP2 interactors. Finally, immunoprecipitations followed by western blot assays confirmed a differential interaction with actomyosin protein MYH10 after expressing *EGFP-PKP2 or EGFP-PKP2-p.R735** in cells (Fig. 1b, c) validating actomyosin components as the major differential interactors between wild-type PKP2 and mutant p.R735* proteins.

### Mutant PKP2 modifies structure of actin filament network

Since active MYH10 binds actin filaments and bundles them to form stress fibers, the crosslinking of actin filaments by MYH10 increases the overall stiffness of actin networks. We hypothesized that PKP2-p.R735* protein modifies actomyosins organization and function, altering F-actin fiber organization and decreasing cellular resistance to deformation (stiffness). We measured cell stiffness in intact HL-1 cardiac cells by acquiring force-volume maps by atomic force microscopy (AFM). Although HL-1 cells derived from atrial cardiomyocytes, they are considered a valuable model to study basic biology of contractile differentiated cardiac cells[23]. The force maps obtained were used to represent spatial variation in Young's modulus as a measure of local cell elasticity[24] in cell lines stably encoding PKP2 or mutant PKP2-p.R735*. The nanomechanical maps show that Young's modulus was higher in regions with a high density of branched actin cytoskeleton networks and stress fibers (Fig. 1d). These observations were quantified by statistical analysis of Young's modulus maps using a bottom-effect correction expression[25] (Supplementary Fig. 1). The graphs show that the median value of the Young's modulus measured in PKP2 cells was higher than that measured in PKP2-p.R735* cells (Fig. 1d). These results support the observation that PKP2-p.R735* cells lack most stress and F-actin fibers.

Biochemically, cells expressing the PKP2-p.R735* mutant had a skewed F-actin:G-actin ratio compared with control PKP2 cells, indicating a shift toward decreased actin polymerization (Fig. 2a). Images also revealed a scarcity of organized F-actin filaments and sarcomere-like structures in the cells encoding the PKP2-p.R735* mutant (Fig. 2b). F-actin filament length quantification using images from cytoplasmic area of HL-1 cells revealed a reduction in length of actin filaments in cells expressing the *PKP2-p.R735** mutant compared with the *PKP2* controls (Supplementary Fig. 2). We also explored the effect of PKP2 and the PKP2-p.R735* mutant proteins on the actin organization of mouse neonatal cardiomyocytes (MNC). The percentage of MNC displaying an organized sarcomeric network varied between transduced cardiomyocytes, with a larger proportion of PKP2 cells showing regions with established sarcomeres compared to mutant PKP2-p.R735* transduced MNC 69.97 ± 6.5% in PKP2 vs. 30.02 ± 6.1% in PKP2-p.R735*; $n = 54$–79; Fig. 2c). To quantify sarcomeric structural configuration in MNC, we used fast Fourier transform of cardiomyocyte images after phalloidin staining[26]. The regularity of sarcomeric striations within the cardiomyocytes resulted in the appearance of peaks in the Fourier spectrum of the image. The frequency of the peak was used to determine the sarcomere period, that was the same (1,78 μm) in both groups (Fig. 2d), and the amplitude of this peak, the sarcomeric power, which can be considered the best indicator of the sarcomere organization level (73.08 ± 12.69 Arb.Units in PKP2 vs. 61.92 ± 6.47 Arb.Units in PKP2-p.R735*; $n = 10$–19; Fig. 2e). The higher the sarcomeric power value, the greater the sarcomere organization. We also calculated the general organization level (coherency) of the actin fiber network in these PKP2 and PKP2-p.R735* cardiomyocytes (Fig. 2f). This parameter of the actin fibers was analyzed to verify the dominant local orientation on the images from PKP2 and PKP2-p.R735* cells using the ImageJ plugin OrientationJ[27] (Supplementary Fig. 3). Images revealed that coherency in control cells was higher compared to the organization found in the actin fibers in PKP2-p.R735* cells, that shows a mostly random organization with values close to 0 (0.37 ± 0.23 Arb.Units in PKP2 vs. 0.13 ± 0.08 Arb.Units in PKP2-p.R735*; $n = 78$–98). Higher coherency values observed in control PKP2 MNC reflected that the direction of the actin fibers in space were forming oriented structures (Fig. 2b, f and Supplementary. 3).

Transverse maximal projections of z-stack images in cells expressing PKP2 confirmed the organization of thick actin filament bundles into a curved structure over the nucleus. In contrast, both in vitro models expressing *PKP2-p.R735** lacked this structure and the nucleus was close to the external plasma membrane (Supplementary

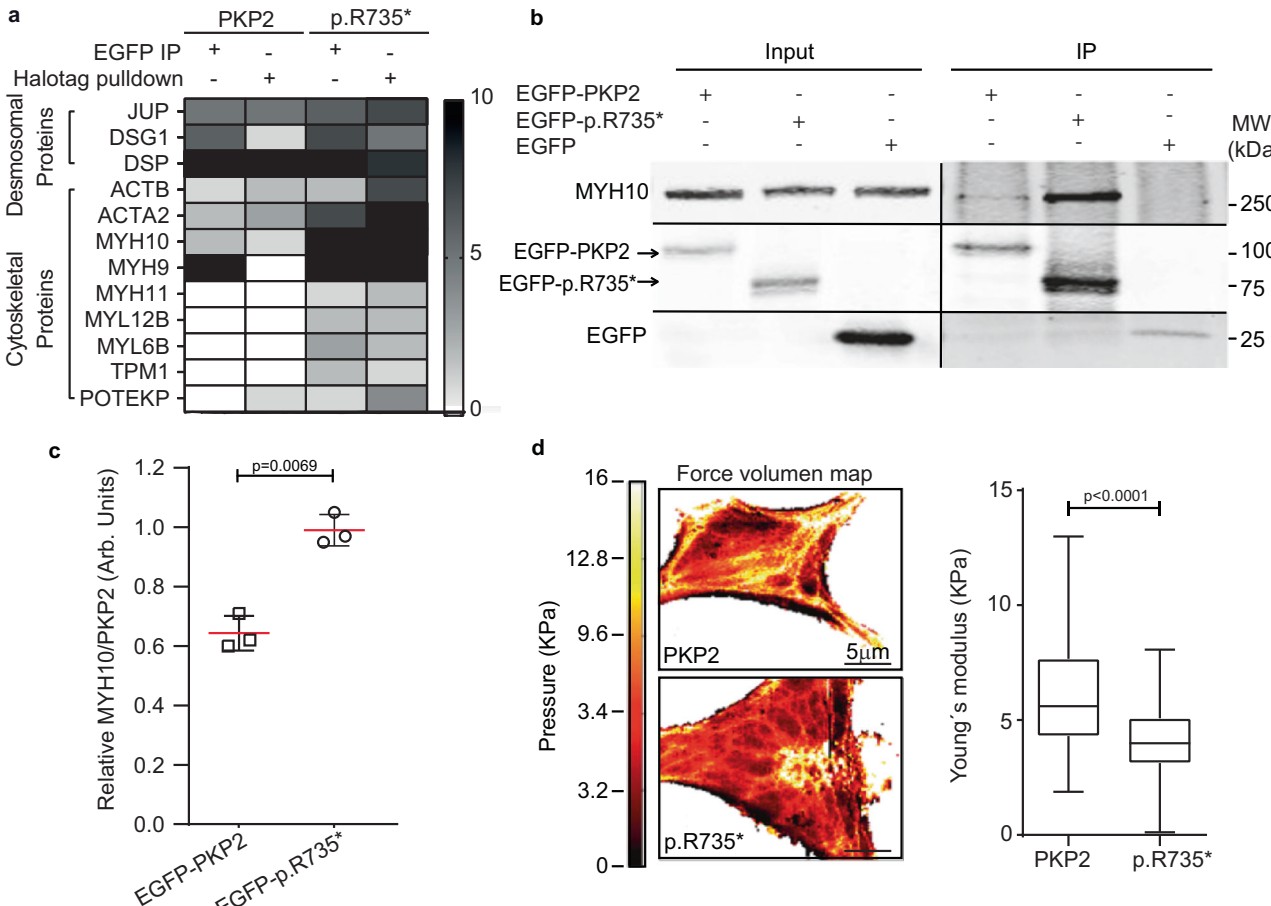

**Fig. 1 | PKP2 interacts with actomyosin proteins and alters actin cytoskeleton.**
**a** Unique peptide counting of most abundant proteins identified in the orthogonal
HaloTag pulldown and EGFP immunoprecipitation assays. The indicated EGFP- or
Halo-tagged proteins were stably expressed in HEK293T cells and subjected to pull-
down or immunoprecipitation with Halo-Link resin or α-EGFP antibodies respectively,
prior to proteomic analysis. The enrichment of cytoskeletal proteins in mutant
p.R735* relative to control PKP2 cells is shown, box displaying the greatest enrich-
ment of proteins known to form the cytoskeleton are indicated in black.
**b** Representative western blot of proteins co-immunoprecipitated with EGFP (28KDa),
EGFP-tagged wild-type PKP2 (120KDa), or EGFP-tagged PKP2-p.R735* mutant
(104KDa) proteins. Blots show PKP2 wild-type and mutant versions, MYH10 or EGFP.

Blots from input samples are also shown. **c** Quantification of MYH10 immunopreci-
pitation using PKP2 and PKP2-p.R735* fusion EGFP proteins as bait. Data are presented
as mean ± sem; $n = 3$ and show relative amount of MYH10 to PKP2 or R735* in the
immunoprecipitation assays. **d** Illustrative atomic force microscopy (AFM) image of
HL1 cells stably expressing *PKP2* or mutant *p.R735** genes. Scale bars, 5 μm. Box plot
showing quantification of Young's modulus values obtained by fitting force-distance
curves. Data are presented from three independent experiments as median with
minimum to maximum values (box limits); $n = 8080$ in PKP2 cells and $n = 93134$ in
PKP2-p.R735* cells, Significance was established as $p < 0.05$, (Mann–Whitney test).
Arb. Units arbitrary units, kPa kiloPascals, MW molecular weight, kDa kiloDalton.
Source data are provided as a source data file.

Fig. 4). This was reflected in the abnormal height of most PKP2-p.R735*
cells, which were shorter than PKP2 cells (5.41 ± 0.24 μm in PKP2 MNC
vs. 4.71 ± 0.21 μm in PKP2-p.R735* MNC; $n = 22$; 4.50 ± 0.22 μm in PKP2
HL-1 cells vs. 3.73 ± 0.27 μm in PKP2-p.R735* HL-1 cells; $n = 13$) (Sup-
plementary Fig. 4). These data show that the higher absolute levels of
F-actin and stress-fiber formation in PKP2 cells are associated with
greater cell stiffness, whereas the lower cell stiffness in mutant p.R735*
correlates with a reduction in F-actin and stress fiber disassembly.

To test whether PKP2-p.R735* functions through Myh10 to disturb
actin filament network, we coexpressed *PKP2-p.R735** and functional
*EGFP-MYH10* genes in HL-1 and MNC. Confocal imaging confirmed that
cells transfected with wild-type *EGFP-MYH10* significantly improved
actin network, reestablishing orientated actin structures (coherency)
in cells expressing the deleterious PKP2-p.R735* mutant (Fig. 3, 3b and
Supplementary Fig. 5). We also detected that expression of *MYH10*
GOF mutant *p.R577** fused to *EGFP* changed the actin cytoskeleton
organization and distribution (Fig. 3c, d). Compared to the diffused
staining observed when mutant EGFP-MYH10-p.R577* is present, EGFP-
Myh10 showed an organized distribution along the F-actin fibers
(Supplementary Fig. 6). Taken together these results support a model

where differential protein-protein interaction in PKP2-p.R735* with
MYH10 alters actomyosin network organization and function.

## PKP2 C-terminal deletion modifies its subcellular localization and defines actomyosin function

As subcellular localization is essential to protein function, we studied
the effect of C-terminal deletion of PKP2 in protein spatial distribution
in MNC and HL-1 cells. Confocal microscopy revealed that most of the
PKP2-p.R735* mutant signal was located in the cytosol, displaced from
the cell edge (Figs. 2b, 4a, 4b and Supplementary. 7). Images and
quantification analysis confirmed that PKP2-p.R735* mutant was pre-
sent at lower levels by the plasma membrane relative to the cytoplasm
(Fig. 4a, b). After transfection with *pCAG-PKP2*, *pCAG-PKP2-p.R735**, or
both plasmids, images and immunoblot analysis of cell fractions
demonstrated that wild-type PKP2 associates with the plasma mem-
brane, whereas the PKP2-p.R735* mutant is predominantly detected in
the cytoplasm and only occasionally at the membrane fraction
(Fig. 4c). Confocal images from MNC encoding PKP2 and PKP2-p.R735*
tagged protein (fused to tdTomato or EGFP respectively) showed that
PKP2 was localized mainly in cell-cell border at the desmosomal plaque

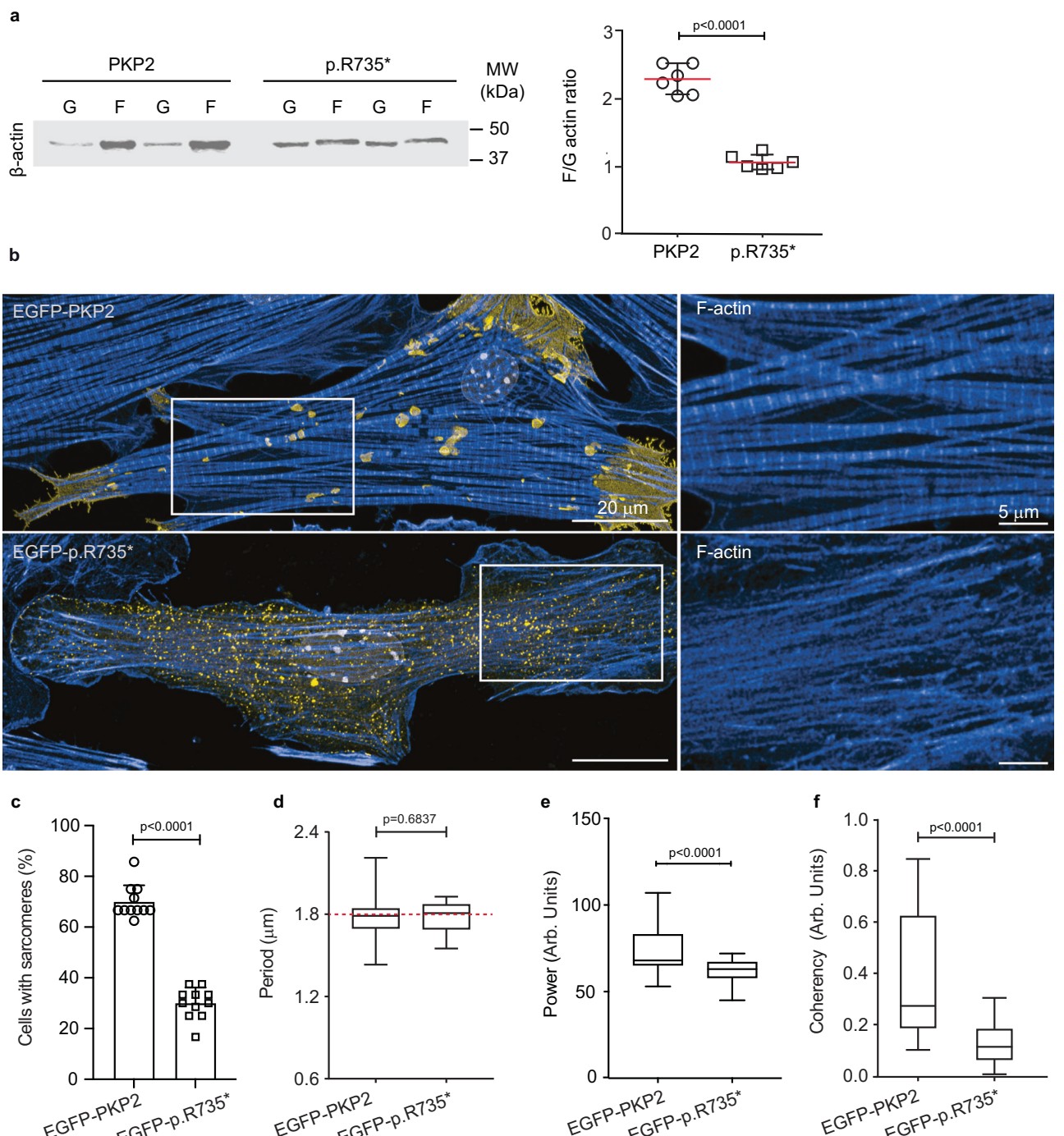

following a linear pattern. In contrast, PKP2-p.R735* was mainly localized at the cytoplasm, and exceptionally at the cell-cell contact area. These data shows that PKP2 protein subcellular localization is not altered when co-expressed with mutant PKP2-p.R735* protein (Fig. 4a–c). Consistently, imaging analysis of heart sections from mice transduced with adeno-associated viruses (AAV) encoding PKP2 or p.R735* EGFP N-terminal fusion protein confirmed the differential cytoplasmic localization of mutant protein (Fig. 4d). These results demonstrate that the C-terminal region of PKP2 is important for proper protein localization at the border contact region in vitro and in vivo. Also, highlight that PKP2 subcellular localization at the desmosome is not altered when the wild-type PKP2 protein is co-expressed with mutant PKP2-p.R735*. These data suggest that the GOF mutant PKP2 has a additional activity in the cytosol, as such

function is not mediated by a deregulation of the levels or proper localization of wild-type PKP2.

In AAV-PKP2-p.R735* animals left ventricle (LV) function is comparable to controls expressing PKP2 although right ventricle end systolic volume (RV-ESV) was significantly larger in AAV-PKP2-p.R735* mice ($28.6 \pm 4.6$ μL in AAV-PKP2-R735* vs. $20.0 \pm 6.1$ μL, and $20.6 \pm 2.4$ μL in non-transduced control mice, and AAV-PKP2 mice respectively; $n = 6$), whereas RV-end diastolic volume (RV-EDV) was similar in all mouse groups. Likewise, RV-ejection fraction (RV-EF) was lower in AAV-PKP2-p.R735* mice ($57.8 \pm 5.3\%$ in AAV-PKP2-R735* mice vs. $71.1 \pm 3.5\%$, and $69.5 \pm 3.8\%$, in non-transduced control mice, and AAV-PKP2 mice respectively; $n = 6$) (Fig. 4e and Supplementary Table 1). These data show that the cardiac expression of an ACM PKP2-related pathogenic mutant causes PKP2 delocalization and a RV systolic dysfunction.

**Fig. 2 | Mutant PKP2 modifies structure of actin filament network.**
**a** Representative immunoblots showing the distribution of β-actin in pellet (F-actin) and supernatant (G-actin) fractions of cells encoding *PKP2*, or *p.R735\** mutant. Graph shows F-actin:G-actin ratios calculated from densitometry analysis. Data are presented as mean ± sd; *n* = 6. Significance was established as *p* < 0.05 (unpaired Student *t* test). **b** Confocal images showing phalloidin-staining of the F-actin network (blue) in mouse neonatal cardiomyocytes (MNC) expressing PKP2 or p.R735\* proteins tagged with EGFP (yellow). Nuclei were stained with DAPI (gray). Magnification shows the different structure of F-actin depending on the expression of PKP2 or p.R735\*. Scale bars, 20 μm and 5 μm. **c** Percentage of MNC with visually apparent and organized sarcomeres; *n* = 54 EGFP-PKP2 cells, *n* = 79 EGFP-pR735\* cells examined over five independent experiments. **d** Sarcomeric periodicity in cells with organized sarcomere network in each group of cells. Red dot line defines normal sarcomeric period (1,78 μm). Boxes depict the 25th–75th percentile with a line showing the median. Whiskers display minimum to maximum values. *n* = 19, ROIs for EGFP-PKP2, *n* = 10 ROIs for EGFP-PKP2-p.R735\*. **e** TTorg analysis of the captured images assessed the organization of the sarcomeres, and this is represented by the

sarcomeric power metric. Calculated transverse organization level of sarcomeres is based on the calculation of the peak amplitude in the Fourier spectrum of the image at the sarcomeric frequency. Boxes depict the 25th–75th percentile with a line showing the median. Whiskers display minimum to maximum values (*n* = 19, ROIs for EGFP-PKP2, and *n* = 10 ROIs for EGFP-PKP2-p.R735\*). **f** Analysis of cellular actin orientation (coherency), using the ImageJ plugin OrientationJ, was performed with an automated routine to quantitatively assess the differences among groups analyzing 20 × 20 μm ROIs from confocal images. Orientation coherency increases with the orientation anisotropy in the image (from zero for random orientations to one for unique orientations). Boxes depict the 25th–75th percentile with a line showing the median. Whiskers display minimum to maximum values; *n* = 78 ROIs for PKP2, *n* = 98 ROIs for PKP2-p.R735\*. Statistical significance was determined by unpaired *t*-test with Welch's correction (two-sided) in (**c**–**f**). Significance was established as *p* < 0.05. Data are presented as mean ± sd. MW molecular weight, kDa kiloDalton, % percentage, μm micrometers, Arb Units arbitrary units. Source data are provided as a source data file.

## PKP2 N-terminal misfolding recovery ameliorates protein delocalization and cardiac dysfunction

To gain insight into how the shortened mutant PKP2 protein affects protein localization and function, we modeled and compared the structures of wild-type PKP2 and the p.R735\* mutant. The best models with minimal energy and correct folding were selected after a final refinement cycle to minimize clashes. Three-D views revealed a major difference in topology and orientation of the N-terminal between wild-type PKP2 and mutant p.R735\* proteins, suggesting that the N-terminal region is a disordered structure. This assumption is also supported by analysis with disopred v3.16, which predicts secondary structure, disordered regions, and protein-binding profiles. Thus, PKP2 sequence analysis predicts two disordered regions, corresponding to the N-terminal and a previously described disordered inter-ARM domain. However, potential misfolding of the PKP2 N-terminal domain is restricted by the C-terminal region, which generates a stable structure. Because of the lack of the C-terminal region in the p.R735\* mutant, the N-terminal domain is unable to fold properly along the central core of ARM repeats to generate a stable structure. In this situation, the N-terminal domain is a disordered region, able to adopt a variable number of conformational states. The reorientation of the N-terminal domain in the p.R735\* mutant alters its topology and exposes an internal section that remains buried in the wild-type PKP2 structure due to the stabilizing action of the C-terminal domain (Fig. 5a). This conformational adjustment also modifies the global electropotential properties at the protein surface, which could explain the altered interaction of p.R735\* mutant PKP2 with partners.

Together, these findings speak to a model wherein the delocalization of PKP2 is a consequence of an unstable conformation that alters MYH10 activity to properly organize the actin filaments. To test whether N-terminal instability is a general feature of truncated mutants of PKP2 we analyzed additional C-terminal mutants reported in the ACM ClinVar database (https://www.ncbi.nlm.nih.gov/clinvar/). We then generated two additional PKP2 C-terminal deletion mutants identified in familiar cases[1], (2028G > A) p.W676\*, and (2421C > A) p.Y807\* fused to an EGFP and expressed in MNC. The two mutants are predicted to alter PKP2 stability and consequently its localization and actin organization (Fig. 5b). In fact, in all mutants there are 2 disordered regions predicted from sequence corresponding to the N-term and a previously described disordered inter-ARM region. The short C-terminal is unable to stabilize the N-domain which is intrinsically unstable. Then, the N-terminus is unable to fold properly along the central core ARM repeats to generate a stable structure, only covering partially the cavity generated by the ARM-repeat domain.

Image analysis revealed that the p.W676\*, p.Y807\* and p.R735\* PKP2 mutants´ signal were in the cytosol, and not associated to the plasma membrane (Fig. 5b). After cell transfection with

*pCAG-EGFP-PKP2*, *pCAG-EGFP-PKP2-R735\**, *pCAG- EGFP-PKP2-p.W676\**, *pCAG-EGFP-PKP2-p.Y807\** confocal imaging also demonstrated that wild-type PKP2 associated with the plasma membrane maintained a well-organized actin cytoskeleton, whereas C-terminal deletion mutants that are predominantly detected in the cytoplasm showed a disorganized cytoskeleton and a lack of organized F-actin filaments (Fig. 5c).

As all these C-terminal deletion mutants show a disorganized N-terminal that exposes the internal region of PKP2, we hypothesized that occlusion of the internal cavity with an unrelated tag could be sufficient to partially rescue the phenotype associated with mutants´ delocalization. As a proof of concept we tested this hypothesis in the PKP2-p.R735\*mutant. We generated a PKP2-p.R735\*-EGFP 'closed' fusion protein, in which the EGFP partially blocks the internal fragment in PKP2-p.R735\* and stabilizes the mutant structure (Fig. 6a). Confocal z-stack images of transfected MNC detected PKP2-p.R735\*-EGFP not only in the cytoplasm but also at higher levels at the plasma membrane, demonstrating partial rescue of proper PKP2 localization (Fig. 6b). Quantitative analysis further proved that PKP2-p.R735\*-EGFP recovered membrane associated localization (Fig. 6c). Confocal imaging of phalloidin staining in cells showed that cells encoding PKP2-p.R735\*-EGFP had a better-conformed actin cytoskeleton with more orientated actin structures than cells expressing unmodified EGFP-p.R735\* (Fig. 6d).

We also tested whether AAV-mediated in vivo expression of the *PKP2-p.R735\*-EGFP* mutant in mouse cardiomyocytes would develop the ACM phenotype associated with cardiac-specific *p.R735\** expression. Unlike AAV-*p.R735\** mutant, AAV-expressed *p.R735\*-EGFP* did not significantly increase RV-ESV (24.5 ± 4.2 μL in *PKP2-R735\**-EGFP mice vs. 21.4 ± 4.8 μL in AAV-PKP2 mice; *n* = 8–10) or significantly reduce RV-EF (64.1 ± 5.6% in AAV-*PKP2-R735\**-EGFP mice vs. 68.8 ± 5.3% in AAV-*PKP2* mice; *n* = 8–10) maintaining similar cardiac function when compared to AAV-*PKP2* (Fig. 6e, f and Supplementary Table 2) and confirming a reduced proficiency to induce ACM in these mice. Western blot analysis of two hearts per group from EGFP tagged proteins (Fig. 6g) revealed similar levels of EGFP signal between AAV-*PKP2-R735\**-EGFP and AAV-*EGFP-PKP2-R735\** animals, confirming that functional differences observed in the MRI analysis are not mediated by differential protein levels between *EGFP-PKP2-R735\** and *PKP2-R735\*-EGFP*. Finally, to examine whether systolic contractile differences observed between these hearts are originated in the cardiac cells itself we tracked sarcomere lengths in isolated cardiomyocytes across the contractile cycle from AAV-*EGFP*, AAV- *EGFP-PKP2*, AAV-*EGFP-PKP2-p.R735\** and AAV-*PKP2-p.R735\*-EGFP*. At relaxed state there were no differences between sarcomere lengths in all tested groups. The analysis of isolated control cardiomyocytes showed that sarcomeric shortening was comparable between untransduced control, and AAV-*PKP2* mice, demonstrating no effect of AAV infection on[28] (Fig. 6h). Isolated cardiomyocytes from

AAV-*EGFP-PKP2-p.R735** mice had decreased maximal shortening compared with controls what demonstrates an effect of PKP2 C-terminal mutant on contractility. When compared with AAV-*EGFP-PKP2*, cardiomyocytes from AAV-*EGFP-PKP2-p.R735** mice had 45% reduced shortening, whilst shortening was partially recovered in cardiomyocytes expressing AAV-*PKP2-R735*-EGFP* (Fig. 6h and Supplementary Fig. 8). All together, these data validate that PKP2 delocalization is critical for protein dysfunction and the ability of

mutant PKP2 to alter sarcomeric contraction and compromise RV cardiac systolic function.

**Mice encoding PKP2-p.R735* or MYH10-p.R577* mutant present sarcomere alterations and RV systolic dysfunction**

It has been recently shown that coordinated functions of MYH9 and MYH10, in time and space, are required for proper sarcomere assembly in an in vitro model of human cardiomyocyte[29]. We questioned

**Fig. 3 | Mutant PKP2 functions through MYH10 to disturb actin organization.**
**a** Representative images of actin filaments (yellow) in MNC expressing tdTomato
fusion PKP2 or PKP2-p.R735* (blue) together with functional EGFP-MYH10
(Magenta). Right panels show a magnification of actin filaments in cells with or
without EGFP-MYH10. This experiment was repeated independently 3 times. Scale
bars, 10 μm and 5 μm. **b** Graphs show quantification from confocal images (**a**) of
actin filament coherency (organization). Boxes depict the 25th–75th percentile with a
line showing the median. Whiskers display minimum to maximum values. ($n = 36$
ROIs for tdTomato-PKP2 transfection; $n = 40$ ROIs for tdTomato-PKP2 with EGFP-
MYH10 cotransfection; $n = 17$ ROIs for EGFP-PKP2-p.R735* transfection; $n = 51$ ROIs
for tdTomato-PKP2-p.R735* with EGFP-MYH10 cotransfection) Statistical sig-
nificance was determined by one-way ANOVA with the Tukey multiple comparison
post-test. Significance was established as $p < 0.05$. Data are presented as mean ± sd.
**c** Representative confocal images showing merge signals of phalloidin-staining of
the F-actin network (blue), wild-type EGFP-MYH10 or mutant EGFP-MYH10-p.R577*
(yellow) and nuclei (gray) in MNC. A magnification of selected white box area
showing F-actin signal. **d** Analysis of cellular actin orientation (coherency), using
OrientationJ, to quantitatively assess the differences among wild-type EGFP-MYH10
or mutant EGFP-MYH10-p.R577* groups. Boxes depict the 25th–75th percentile with a
line showing the median. Whiskers display minimum to maximum values. $n = 24$
ROIs for MYH10; $n = 37$ ROIs for MYH10-p.R577*. Statistical significance was
determined by unpaired $t$ test with Welch's correction (two-sided). Significance was
stablished as $p < 0.05$. Data are presented as mean ± sd. Arb. Units arbitrary units,
μm micrometers. Source data are provided as a source data file.

whether MYH10 disfunction could affect general sarcomere organi-
zation and be responsible for decreased performance in mice
observed by MRI (Fig. 4e). Actin filaments from AAV-*EGFP-MYH10-
p.R577**, AAV-*EGFP-PKP2*, AAV-*EGFP-PKP2-R735**, and non-transduced
sham control mice were visualized in isolated cardiomyocytes by
staining with phalloidin (Fig. 7a) to measure the angle of deviation
from the main direction (striation orientation) that the sarcomeres
follow. Image analysis revealed a significant increase in the AAV-*EGFP-
MYH10-p.R577** and AAV-*EGFP-PKP2-R735** angle variation compared
with the other two groups (5.57 ± 0.18° and 5.56 ± 0.19° in *PKP2-p.R735**
and *MYH10-p.R577** cardiomyocytes respectively vs. 3.69 ± 0.13° and
3.84 ± 0.13° in non-transduced, AAV-*PKP2*; $n = 28$) (Fig. 7b). As less
variation in sarcomere orientation corresponds to optimal force
generation[30], this data predicts a comparative and similar force loss
only in sarcomeres of AAV-*p.R735** and AAV-*EGFP-MYH10-p.R577**
hearts, in agreement with cardiac functional MRI results (Fig. 7c, d and
Supplementary Fig. 9). Consistently, the contraction analysis of AAV-
*EGFP-PKP2-p.R735** and AAV-*EGFP-MYH10-p.R577** isolated cardiomyo-
cytes also showed that sarcomeric shortening was compromised in
both mice lines (Fig. 7e), establishing a clear correlation between sar-
comeric misalignment and reduced systolic contraction in vivo. Alto-
gether, our data shows that PKP2 cytoplasmic delocalization and
MYH10 protein–protein interaction alters actomyosin component
relative distribution and activity, affecting sarcomere organization,
contraction capacity and RV cardiac systolic function inducing an
ACM-like phenotype.

## Activation of MYH10 corrects PKP2-p.R735* defects
As genetic overdose of MYH10 normalized the actomyosin structure of
PKP2-p.R735* cells and dominant negative EGFP-MYH10-p.R577*
mutant mimics contractile defects observed in ACM, we hypothesized
that 4-hydroxyacetophenone (4-HAP), a selective non-muscle myosins
activator, would reduce deleterious effects of PKP2 mutant. This small
molecule activates MYH10 and MYH14 by promoting their assemblies[31]
and has been proposed for treating different types of metastatic
tumors[32,33]. Using in vitro and in vivo models of ACM described above
we studied the response to 4-HAP treatment. Cardiomyocytes
encoding the PKP2-p.R735* mutant had a disorganized actin cytoskele-
ton (Fig. 2b). Images from cytoplasmic area of MNC expressing the
*PKP2-p.R735** mutant after 4-HAP administration (50 μM) revealed an
improved F-actin filaments network (Fig. 8a). We detected a significant
reorganization of PKP2-p.R735* mutant sarcomeres in treated MNC, as
indicated by the increase of cells with evident sarcomeres from 30.02
to 70.83% after 4-HAP treatment (Fig. 8b), and the improvement in
sarcomeric power compared to PKP2 transduced control (61.9 ± 6.4
Arb. Units in untreated PKP2-p.R735* vs. 73.1 ± 12.6, 67.9 ± 12.3 and
70.2 ± 11.2 in non-treated PKP2, and 4-HAP treated PKP2, and mutant
PKP2-p.R735* MNC respectively; $n = 16$–22) (Fig. 8c). We also observed
a significant improvement in the PKP2-p.R735* mutant actin general
spatial disposition, reestablishing orientated actin structures of fibers
after 4-HAP treatment (Fig. 8d). Furthermore, in vivo, administration of
4-HAP (1 mg/kg) for one week improved cardiac RV systolic function

(Fig. 9a) (22.2 ± 6.3 μL in untreated mice vs. 15.9 ± 3.2 μL in 4-HAP
treated AAV-*R735** mice; $n = 8$–12) and general RV performance
(49.1 ± 7.5% vs. 58.5 ± 4.0% in sham and treated AAV-*PKP2-p.R735**-EGFP
mice respectively; $n = 8$–12) (Fig. 9b) in animals expressing *PKP2-
p.R735** mutant. Unlike AAV-*EGFP-PKP2-p.R735** mutant, 4-HAP admin-
istration did not significantly reduced RV-ESV or significantly increased
RV-EF in AAV-*EGFP-PKP2* (Fig. 9 and Supplementary Fig. 10). Con-
sistently, in isolated cardiomyocytes from AAV-*EGFP-PKP2*, and AAV-
*EGFP-PKP2-p.R735** mice after treatment the actin filaments stained
with phalloidin showed that the angle of sarcomeric deviation from the
main direction was restored to normal (Fig. 9e). Sarcomeric realign-
ment in treated mice also correlated with improved contraction
measured in isolated cardiomyocytes. In these cardiomyocytes sar-
comeric shortening was indistinguishable between AAV-*EGFP-PKP2*,
AAV-*EGFP-PKP2-p.R735** after 4-HAP treatment (Fig. 9f). In summary,
therapeutic activation of MYH10 corrects cellular contractile pheno-
type and may potentially limit RV adverse clinical complications in
patients with ACM.

## Discussion
Herein, we reveal that PKP2 C-terminal deletion mutants act as a key
regulator of cardiac contractile disfunction in arrhythmogenic cardi-
omyopathy (ACM) and that their direct protein-protein interactor
MYH10 has a critical role in the pathological process. Four major
findings from our in vitro and in vivo experimental models support this
conclusion. First, expression of C-terminal deletion of PKP2 modifies
MYH10 activity, altering F-actin fiber organization and decreasing
cellular resistance to deformation (stiffness). Second, cardiac-specific
PKP2 C-terminal GOF mutants, contributed to pathological myocardial
ACM by altering the actomyosin network, what translates into sarco-
meric abnormal organization and right ventricular (RV) cardiac func-
tion. Third, mutant *MYH10-p.R577** expression compromises
sarcomere fine tune organization and cardiac performance alike PKP2
C-terminal deletion mutant. Finally, therapeutic administration of
MYH10 activator 4-HAP to animals with reduced RV ejection fraction
(EF) improves systolic function and restored mechanical performance.

Apart from extensive literature on MYH10 role in vitro and during
development[17,34–37], little is known about its function in adult cardiac
cells. It has been shown that MYH10 dysregulation in iPS-derived car-
diomyocytes is associated to increased sarcomere misorientation and
sarcomeric dysfunction. In our in vivo setting, it is possible that
reduced force generated in C-terminal deletion mutants is transduced
into sarcomeres and derived from integrated fiber misorientation has a
consequence on tissue performance. From a biophysical perspective,
vectorial nature of force predicts that an increase in the actin angle
deviation from the perpendicular sarcomere axis will produce a
reduction on the final magnitude of the generated force. Thus, mis-
alignment in a single PKP2 or MYH10 mutant cardiomyocytes can
partially justify a decline in the force the heart muscle can exert to
neighbor sarcomere. Consistently, the image analysis of angle varia-
tion from individual cardiomyocytes correlates with a significant dif-
ference between mutants *PKP2-p.R735** and *MYH10-p.R577**, with

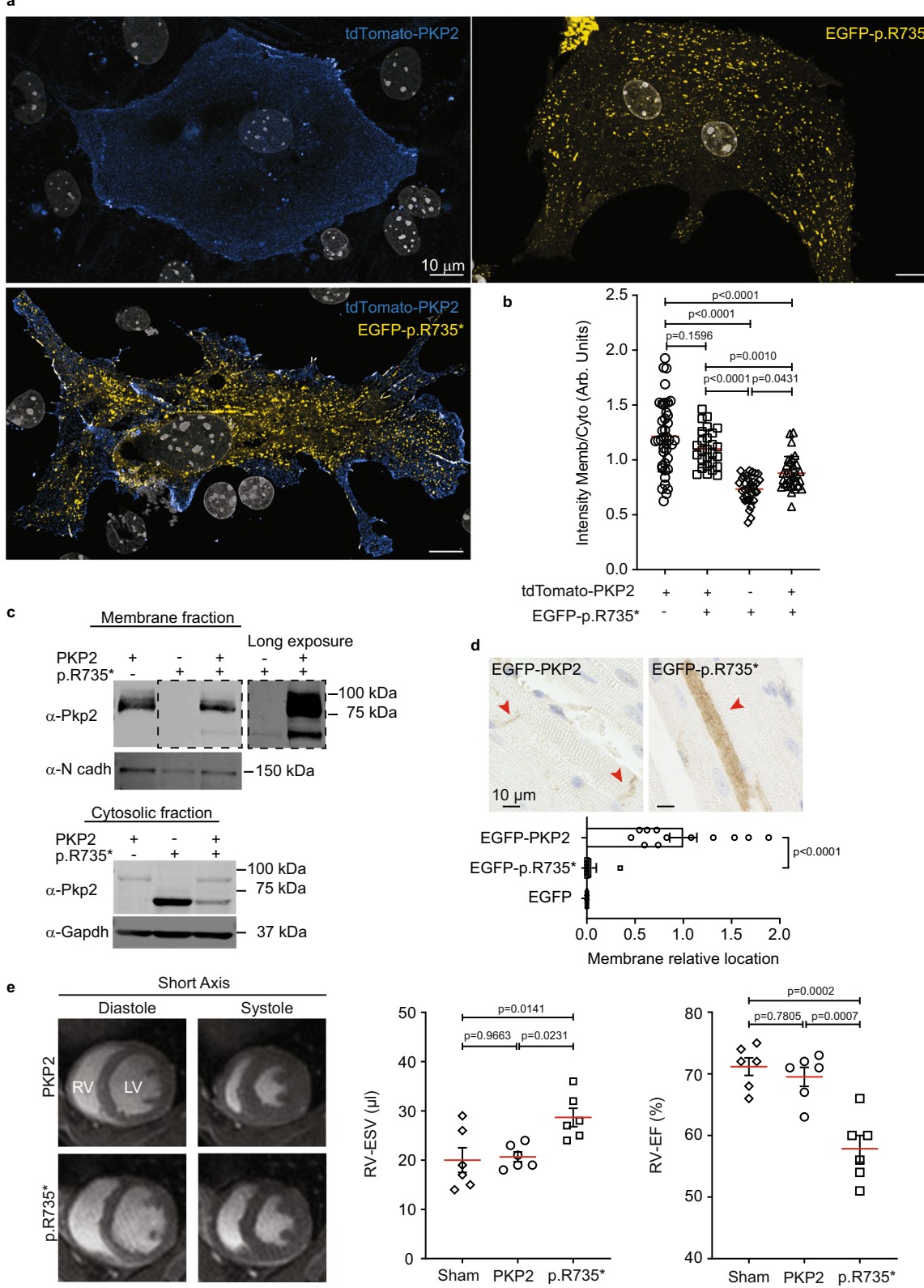

transduced control animals, not observed either in wild-type mice or animals treated with 4-HAP for one week. We speculate that such variation in the sarcomeric organization of an ultrastructure-related force is at least in part behind the deficit of heart muscle contraction observed in functional MRI analysis (Figs. 4, 6, 7 and 9).

As mentioned above, dominant inherited cardiomyopathy could be caused by loss-of-function (haploinsufficiency), a dominant negative effect or GOF. We identified a possible mechanism through which the PKP2-p.R735* variant could function as a GOF mutation and not as a dominant negative or a simple loss-of-function allele induced by a nonsense-mediated mRNA decay mechanism. As proteins need to be in the correct cellular compartment to perform their function, it is not surprising disease mutations can lead to depletion of the protein from its native location, or its accumulation in the wrong

**Fig. 4 | PKP2 C-terminal deletion modifies its subcellular localization and defines actomyosin function. a** Representative images of showing the distribution in mouse neonatal cardiomyocytes (MNC) of tdTomato-PKP2, EGFP-PKP2-p.R735* mutant or both. Scale bars, 10 μm. **b** Dot-plot showing the distribution of PKP2 versions as ratio intensity of membrane-cytoplasm. Lanes one and two show PKP2 membrane-cytoplasm ratio intensity, last two lanes show PKP2-p.R735* ratio; *n* = 41 cells for tdTomato-PKP2 tranfection, *n* = 27 cells for *TtdTomato-PKP2* with *EGFP-PKP2-p.R735** cotransfection; *n* = 33 cells for EGFP-PKP2-p.R735* transfection; *n* = 31 cells for EGFP-PKP2-p.R735* with tdTomato-PKP2 cotransfection from three independent experiments. Significance was stablished as *p* < 0.05 (one way ANOVA with Tukey multiple comparison post-test). Data are presented as mean ± sd. **c** Immunoblot analysis of membrane and cytosolic cell fractions. Symbols + and – indicate transfection or not with vectors expressing *PKP2, p.R735**, or both variants together. Blots show PKP2 (wild-type and mutant versions), N-cadherin (load control of membrane fraction) and GAPDH (load control of cytosolic fraction). Black dotted lines delimit short and long exposures of the same blot to detect p.R735* and PKP2 proteins. The results shown are representative of three independent experiments. **d**, Images of heart sections immunostained with EGFP antibody. Animals were transduced with AAV-*EGFP*, AAV-*EGFP-PKP2* or AAV-*EGFP-PKP2-p.R735**. Red arrowheads indicate PKP2 or PKP2-p.R735* cellular location. Bottom graph shows relative percentage of membrane localization; *n* = 9 mice (EGFP and EGFP-PKP2), and *n* = 7 mice (EGFP-PKP2-p.R735*). Scale bars, 10 μm. **e** Representative short-axis cardiac MRI images taken at the end of diastole and systole in mice transduced with wild-type AAV for *PKP2* or the *p.R735** mutant. Quantification of right ventricular (RV) end systolic volume (RV-ESV) and ejection fraction (RV-EF), by MRI in anesthetized non-transduced sham control mice, AAV-*PKP2* and AAV-*PKP2-p.R735** transduced mice. Data are presented as mean ± sd; *n* = 6 mice. Statistical significance was determined in (**d**, **e**) by one-way ANOVA with Tukey multiple comparison post-test. Significance was established as *p* < 0.05. Data are presented as mean ± sem. Arb. Units arbitrary units, KDa kiloDalton, μm micrometers, μL microliters, % percentage, RV right ventricle, ESV end systolic volume, EF ejection fraction. Source data are provided as a source data file.

compartment. We show that mutant PKP2-p.R735* is mislocalized, although does not sequester or alter wild-type PKP2 localization and dynamics at the cell-to-cell contact area (Fig. 4). This data suggests that PKP2-p.R735* mutant effect is not dependent on wild-type PKP2 dysfunction or cellular mislocalization and must be a consequence of acquired new activities. We determined that the pathogenic PKP2-p.R735* variant functions by interrupting or rewiring highly connected interaction networks to disturb F-actin homeostasis. We therefore predict that GOF is the mode of action of other C-terminally truncated PKP2 variants stable to retain MYH10 interaction and to disrupt actomyosin organization. The conformational changes induced by C-terminal deletion mutations in PKP2 alter mutant proteins localization and function (Fig. 5). PKP2 C-terminal deletions exposes the normally buried internal interface to the solvent. This structural alteration in PKP2 may rewire highly connected cellular interaction networks in a subtle but disease-specific manner. Protein trafficking and localization is regulated by interplay between diverse cis-acting targeting sequences and trans-acting trafficking factors. Thus, PKP2 C-terminal deletions may affect protein trafficking and localization either directly, through the loss of a domain essential for directing PKP2 to the desmosome, or indirectly, by generating unknow or altered interactions that keep the protein away from the cell-to-cell contact area. Our results (Fig. 6) make a strong case for this indirect effect, since incorporation of the EGFP tag in the closed PKP2-p.R735*-EGFP mutant partially restored proper protein localization compared with the open PKP2-p.R735*mutant. This experiment also suggests that the PKP2 C-terminal domain lacks a specific motif required for desmosomal localization and that the PKP2 mutants with open conformation generate or strengthen interactions that retain the protein in the soluble fraction.

The ClinVar database (https://www.ncbi.nlm.nih.gov/clinvar/), part of the NCBI Entrez system, attempts to establish relationships between gene variants and phenotype. Among them, 34 are nonsense mutations and 23 span different portions of the 8 armadillo domains in PKP2. Remarkably, all nonsense *PKP2* variants are classified as pathogenic or likely pathogenic, underlining the functional importance of the PKP2 C-terminal domain. We predict that PKP2 mutants that lack the ability to stably localize in the desmosome will share similarities in their phenotypes when they maintain their interaction with MYH10.

The RV and the LV free myocytes are developed from different progenitor cells, their walls possess distinct architecture, and response mechanisms against stress. RV has a thinner free wall and support an intense stress load compared with the LV. The most common subtype of ACM is arrhythmogenic right ventricular cardiomyopathy (ARVC), which is overrepresented in patients with mutations resulting in premature termination of the PKP2 protein[38,39]. Therefore, it is not surprising that pathogenic mutations in PKP2 have been usually associated with the classical form of the disease that predominantly shows

structural abnormalities that lead to a progressive global RV dysfunction[1]. We observed a clear structural and functional phenotype in the RV meanwhile significant functional variations are not evident in the LV of mutant animals. In patients, these phenotypic differences between ventricles have been sometimes explained due to environmental stressors like exercise[16,40] that define the final disease outcome. For example, under extreme exercise conditions the RV goes through a greater load increase compared with the left ventricle, where near-linear increase in pulmonary artery pressures predominantly contributes to a disproportionate increase in RV wall stress[41]. It is reasonable that ACM develops preferentially in the RV in *PKP2* mutant carriers with compromised structural integrity of the myocardial cytoskeleton and half the amount of functional PKP2 protein at desmosomal structures.

Sarcomere assembly, maturation and maintenance is a complex process that depends on multiple components[42,43] Adult cardiomyocytes become isotropic, organizing their contractile cytoskeleton and remodeling nuclei, and junctions[44]. Here we present compelling evidence that mutations in desmosomal protein PKP2 related to ACM disturb sarcomeric homeostasis by modifying MYH10 activity. Our results highlight the role of MYH10 as a key component for sarcomere maintenance in adult cardiomyocyte, including evidence of cardiomyopathy development with their alteration, indicative of sarcomere organization interference.

With the lack of treatments for patients with ACM, our data identifies the actomyosin network in the context of cardiac mechanical performance regulator in ACM disease and highlights the therapeutic potential of MYH10 pharmacological activation to improve sarcomere organization and to recover cardiac contractability. It is essential to understand the pathological mechanism of the target mutation on PKP2 to develop the potentially most beneficial treatment.

## Methods
### Animal
All animal procedures followed the guidelines from Directive 2010/63/EU of the European Parliament on the protection of animals used for scientific purposes. Animal experiments were carried out in accordance with the CNIC Institutional Ethics Committee recommendations and were approved by the Animal Experimentation Committee (Scientific Procedures) of Comunidad de Madrid (project number PROEX 019/17).

All mice used in this study were generated on a C57BL/6J background and were provided by Charles River Laboratories. Three to 5-month-old male mice and neonatal mice of both sexes were analyzed. The animals were maintained in a specific pathogen-free animal facility under controlled temperature (22° ± 0.8 °C), 55 ± 10% relative humidity and a 12 h light/dark cycle. Mice had access to food (Teklad global rat/mouse chow, Harlan Interfauna) and water ad libitum. Adult mice were

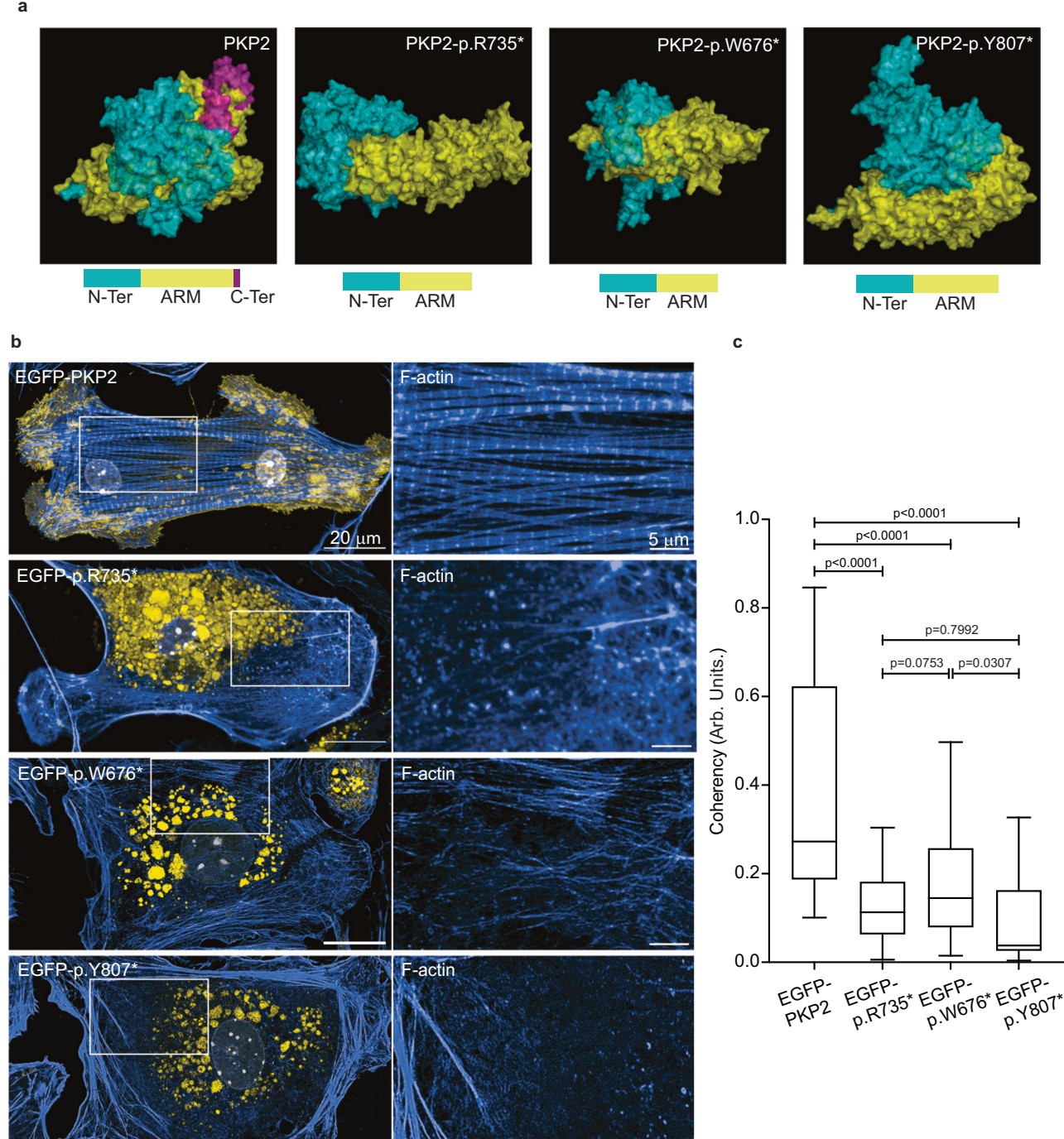

**Fig. 5 | PKP2 C-terminal deletion mutants show a disorganized N-terminal and altered subcellular localization. a** 3D view of protein PKP2, PKP2-p.R735*, PKP2-p.W676*, and PKP2-p.Y807*. Below the schematic diagram of the domains within PKP2 proteins. Different domains are shown: N-domain (N-Ter, blue), armadillo domain (ARM, yellow), C-domain (C-Ter, Purple). **b** Representative images of mouse neonatal cardiomyocytes (MNC) transfected with *EGFP-PKP2, EGFP-PKP2-p.R735*, EGFP-PKP2-p.W676** or *EGFP-PKP2-p.Y807** (yellow) showing its location. Cells are also stained with phalloidin (blue) and DAPI (gray). Scale bars, 20 μm and

5 μm. **c** Analysis of cellular actin orientation, using the ImageJ plugin OrientationJ. Boxes depict the 25th–75th percentile with a line showing the median. Whiskers display minimum to maximum values. $n = 78$ ROIs for EGFP-PKP2 $n = 98$ ROIs for EGFP-PKP2-p.R735*; $n = 49$ ROIs for EGFP-PKP2-p.W676* and $n = 38$ ROIs for EGFP-PKP2-p.Y807*. Statistical significance was determined by one-way ANOVA with Tukey's multiple comparison post-test. Significance was stablished as $p < 0.05$. Data are presented as mean ± sd. Arb. Units arbitrary units, μm micrometers. Source data are provided as a source data file.

euthanized by carbon dioxide inhalation, and neonates P1 by decapitation with sharp well-maintained scissors.

### Adeno-associated virus (AAV) animal models

To develop all animal models, AAV vectors were produced with the double transfection method by using HEK293T (ATCC, CRL-3216) cells

as described previously[16,45]. AAV plasmids were cloned and propagated in the Stbl3 *E. coli* strain (Life Technologies). Shuttle plasmids *pAAV-PKP2, pAAV-PKP2-p.R735**[46], *pAAV-EGFP-PKP2, pAAV-EGFP, pAAV-EGFP-PKP2-p.R735*, pAAV-PKP2-p.R735*-EGFP,* and *pAAV-EGFP MYH10-p.R577** were packaged into AAV-9 capsids with the use of the helper plasmid *pDG9* (providing the three adenoviral helper genes, and rep and cap

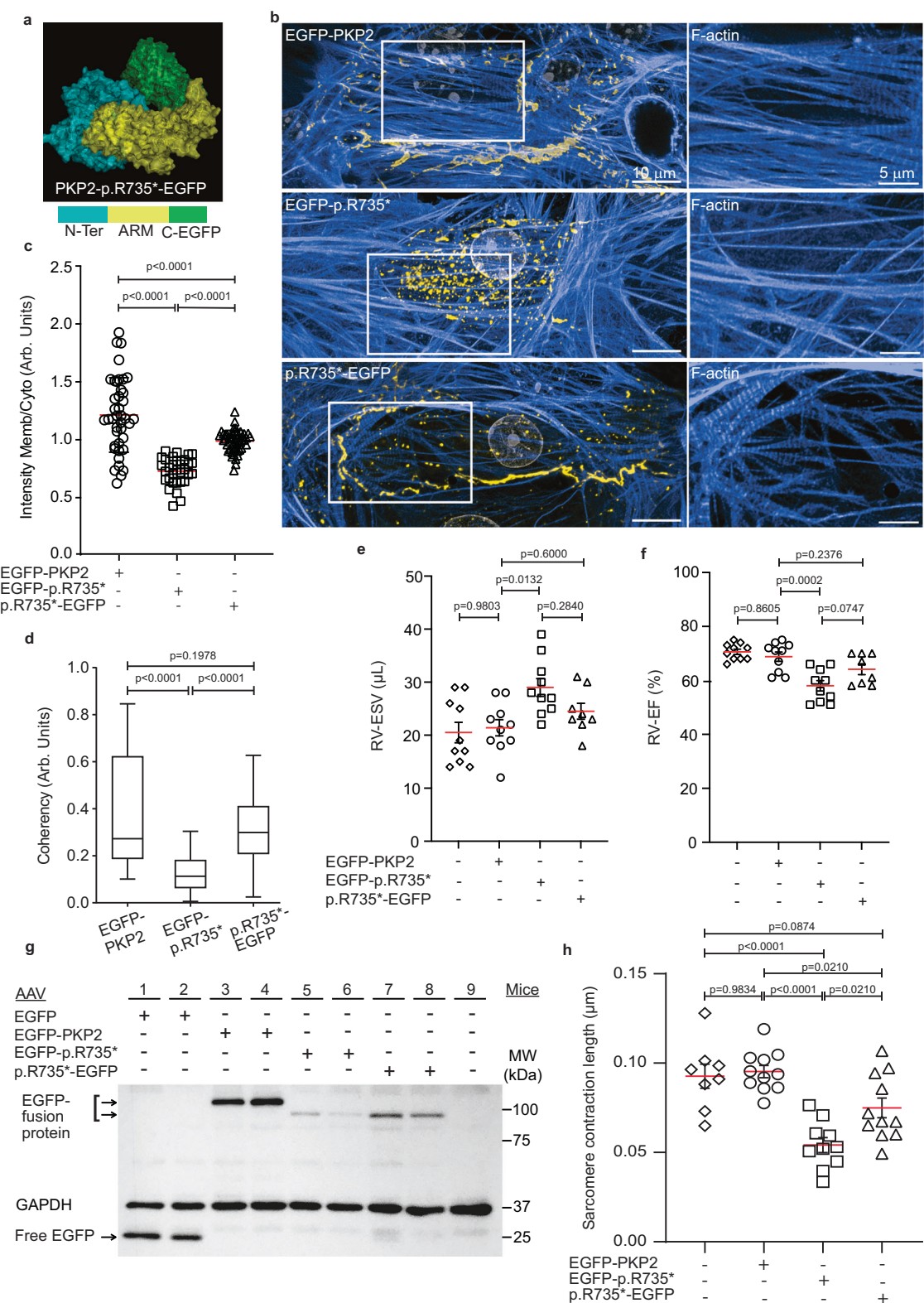

viral genes), obtained from PlasmidFactory. The AAV shuttle and helper plasmids were transfected into HEK293T cells by calcium-phosphate co-precipitation. A total of 690 μg of equimolar-mixed plasmid DNA was used per Hyperflask (Corning) seeded with $1.2 \times 10^8$ cells the day before. Seventy-two hours after transfection, the cells were collected by centrifugation, and the cell pellet was resuspended in TMS (50 mM Tris-HCl, 150 mM NaCl, 2 mM MgCl₂) on ice before digestion with Benzonase nuclease (25 KU/mL; Millipore) at 37 °C for

30 min. Clarified supernatant containing the viral particles was obtained by iodixanol gradient centrifugation[47]. Virus-containing gradient fractions were concentrated through Amicon UltraCel columns (Millipore) and stored at −80 °C. For AAV injection mice were anesthetized with a 100 μL intraperitoneal injection containing ketamine (60 mg/kg), xylazine (20 mg/kg), and atropine (9 mg/kg). Once anesthetized, animals were placed on a heated pad at 37 ± 0.5 °C to prevent hypothermia. A small incision (4 mm) was made to expose the right

**Fig. 6 | PKP2 N-terminal misfolding recovery ameliorates protein delocalization and cardiac dysfunction. a** 3D view of protein PKP2-p.R735*-EGFP with schematic diagram of the domains: N-terminal domain (N-Ter, blue), armadillo domain (ARM, yellow), C-terminal EGFP (C-EGFP, green). **b** Representative images of EGFP-PKP2, EGFP-PKP2-p.R735* and PKP2-p.R735*-EGFP protein localization and F-actin structure in MNC. A magnification of selected white box area showing F-actin signal. Scale bars, 10 μm and 5 μm. **c** Plot shows the quantification of plasma membrane intensity levels normalized to cytoplasm intensity ($n = 41$ cells for EGFP-PKP2 $n = 32$ cells for EGFP PKP2-p.R735* $n = 48$ cells for PKP2-p.R735*-EGFP) from three independent experiments, **d** Analysis of cellular actin coherency from (**b**) ($n = 78$ ROIs for EGFP-PKP2; $n = 98$ ROIs for EGFP-PKP2-p.R735*; n = 31 ROIs for PKP2-p.R735*-EGFP). **e, f** Right ventricular (RV) end systolic volume (ESV) and ejection fraction (EF) determined by MRI in hearts of non-transduced (sham control) and AAV-*EGFP-PKP2*, AAV-*EGFP-PKP2-p.R735**, or AAV-*PKP2-p.R735*-EGFP* transduced mice; ($n = 10$ mice for all the groups but PKP2-p.R735*-EGFP $n = 8$). **g** Mice were transduced as in (**e**), prior to Western blot analysis. Heart lysates were immunoblotted to analyze free EGFP and fusion protein levels. GADPH was used as loading control. **h** Graph showing the difference between the maximum and minimum sarcomere length in ten consecutive beats from isolate adult cardiomyocytes of sham control non-transduced, AAV-*EGFP*, AAV-*EGFP-PKP2*, AAV-*EGFP-PKP2-p.R735** and AAV-*PKP2-p.R735*-EGFP* mice; ($n = 8$, $n = 11$, $n = 10$ and $n = 11$ cardiomyocytes respectively, from three independent mice). Statistical significance was determined by one-way ANOVA with Tukey's multiple comparison post-test. Significance was stablished as $p < 0.05$. Data are presented as mean ± sd (**c, d**) or mean ± sem (**e, f, h**). μm micrometers, Arb. Units arbitrary units, μL microliters, MW molecular weight, kDa kiloDalton, RV right ventricle, ESV end systolic volume, EF ejection fraction. Source data are provided as a source data file.

femoral vein. To increase vessel diameter and facilitate infusion, blood flow was interrupted with a cotton bud for a couple of seconds. Once the vein was dilated, an insulin syringe vessel was introduced into the vein, and $1 \times 10^{11}$ virus particles were injected in a volume of 50 μL, taking care to prevent introduction of air bubbles. Animals were then analgesized with buprenorphine (subcutaneous 0.1 mg/kg) and maintained on the heated pad until recovery. Mice were individually housed in wire-bottomed cages and paracetamol was administered orally for 1 week.

### Cardiomyocyte isolation
Mice were anesthetized by intraperitoneal injection of ketamine and xylazine, and hearts were rapidly excised, cannulated with a 21-gauge gavage feeding needle, connected to a Langendorff perfusion apparatus, and perfused at 37 °C for 5 min at 0.8 mL/min with a perfusion buffer containing (in mM) 113 NaCl, 4.7 KCl, 0.6 $KH_2PO_4$, 12 $NaHCO_3$, 0.6 $Na_2HPO_4$, 0.032 phenol red, 10 $KHCO_3$, 10 HEPES, 30 taurine, 1.2 $MgSO_4 \cdot 7H_2O$, 5.5 glucose, and 10 2,3-butanedione monoxime (BDM), pH 7.46. After 5 min, the hearts were perfused for 20 min at 0.8 mL/min with digestion buffer, consisting of the perfusion buffer plus 12.5 μM $CaCl_2$, 0.14 mg/mL trypsin (Invitrogen, Carlsbad, CA), 0.005 U/ml DNAse (Sigma) and 0.2 mg/mL Liberase Blendzyme 4 (Roche Diagnostics). Hearts were then cut from the cannula below the atrium and placed in a dish containing 2.5 mL digestion buffer. The tissue was gently dissected into small pieces with fine forceps and dissociated by passing through a sterile plastic transfer pipette. The cell suspension was filtered through a 200 μM nylon filter in a 15 mL falcon tube, and digestion was inactivated by addition of 10 mL of first cardiomyocyte stopping digestion buffer (perfusion buffer containing 10% fetal bovine serum and 12.5 μM $CaCl_2$). Cardiomyocytes were allowed to settle to the bottom of the tube for 30 min in darkness. The supernatant was discarded, and the pellet was resuspended in 8 mL of a second cardiomyocyte stopping buffer (perfusion buffer containing 5% fetal bovine serum and 12.5 μM $CaCl_2$). The cells were again allowed to settle to the bottom of the tube for 30 min in darkness, and the supernatant was discarded. The cells were then passed through a series of 20 min incubations in 8 mL of modified second cardiomyocyte stopping buffer with incremental increases in $CaCl_2$ to a final concentration of 1 mM.

### Cardiomyocytes immunostaining and analysis
For phalloidin staining, cardiomyocytes were fixed for 20 min in 4% paraformaldehyde (PFA). Cells were blocked and permeabilized for 1 h in TSA Blocking Reagent (FP1012, PerkinElmer) plus 0.1% triton (T9284, Sigma) at room temperature. Then Phalloidin-iFluor 594 Reagent (ab176757, Abcam) was added at 1:1000 dilution and incubated for 90 min. Finally, after wash with PBS, samples were mounted in Mowiol mounting medium (Mowiol 4-88, Glycerol, 200 mM Tris-HCl pH 8.5 and 2.5% 1,4-diazabicyclo-[2,2,2]-octane). Fluorescence images were obtained with a Leica SP8 confocal microscope with HC PL APO

100x/1.4 oil objective. Actin filament dispersion in adult cardiomyocytes were analyzed using Directionality plugin of Fiji.

### Cardiomyocytes contraction analysis
Cardiomyocyte contraction was recorded using Leica DMi8 with HC PL APO 40× objective and DFC3000G-0047012016 camera. Myocytes were superfused with Tyrode's solution containing the following (in mM): 136 NaCl, 5.4 KCl, 0.9 $CaCl_2$, 1.2 $KH_2PO_4$, 5 HEPES, 10 glucose, at pH 7.4, adjusted with NaOH. Myocytes were electrically field stimulated at 1 Hz with square-wave pulses (9,9 mA, 3 ms) delivered through a pair of platinum electrodes using an Axopatch 200B amplifier. Screen was recorded with OBS program at 80fps. Then contraction was analyzed with Sarcoptim plugin from Fiji software. Ten beats from each cardiomyocyte were analyzed. The difference between the average of maximum and minimum is represented. At least ten cardiomyocytes from three different animals were analyze.

### Cardiac magnetic resonance imaging (MRI)
Cardiac magnetic resonance imaging (MRI) images were obtained from isofluorane- anesthetized animals (induction: 2 vol.% inhaled isoflurane in 1000 $cm^3$/min $O_2$, during measurement: 1.8–1.4 vol.% inhaled isoflurane in 500 $cm^3$/min $O_2$) while monitoring for core body temperature, cardiac rhythm, and respiration rate. In vivo cardiac images were acquired using an Agilent VNMRS DD1 7T MRI system (Santa Clara, California, USA) and a k-space segmented ECG-triggered cine gradient-echo sequence. After shimming optimization, cardiac four-chamber and left two-chamber views were acquired and used to plan the short axis sequence. Mice were imaged with the following parameter settings: number of slices, 14; slice thickness, 0.8 mm; slice gap, 0,2 mm; matrix size, 256 × 256; field of view, 30 × 30 $mm^2$; gating, ECG and respiratory triggered; cardiac phases, 20; averages, 4; Echo-Time -1.25 ms, minimum TR, 6 ms; flip angle, 15°; trigger delay, 0 ms; trigger window, 5 ms; dummy scans, 2. Images from short axis cine gradient-echo MRI sequences were quantitatively analyzed by manual detection of endocardial borders in end-diastole and end-systole, with exclusion of papillary muscles.

### Cell lines
The HEK293T (ATCC, CRL-3216) cell line was maintained in DMEM (GIBCO) supplemented with 10% FBS, 1% penicillin/streptomycin, and 2 mM L-glutamine. The atrial cardiomyocyte cell line HL-1 (Sigma, Aldrich) was maintained in Claycomb medium (Sigma Aldrich) supplemented with 10% FBS, 1% penicillin/streptomycin, and 2 mM L-glutamine, as previously described[48]. HL-1 cells were seeded on plates coated with 0.02% gelatin/fibronectin (Sigma Aldrich). Cell lines were maintained at 37 °C with 5% of $CO_2$.

### Mouse neonatal cardiomyocyte (MNC) isolation
Cardiac cells were isolated from day-old neonatal mouse hearts, previously segregated by sex[49], in sterile conditions as described in the

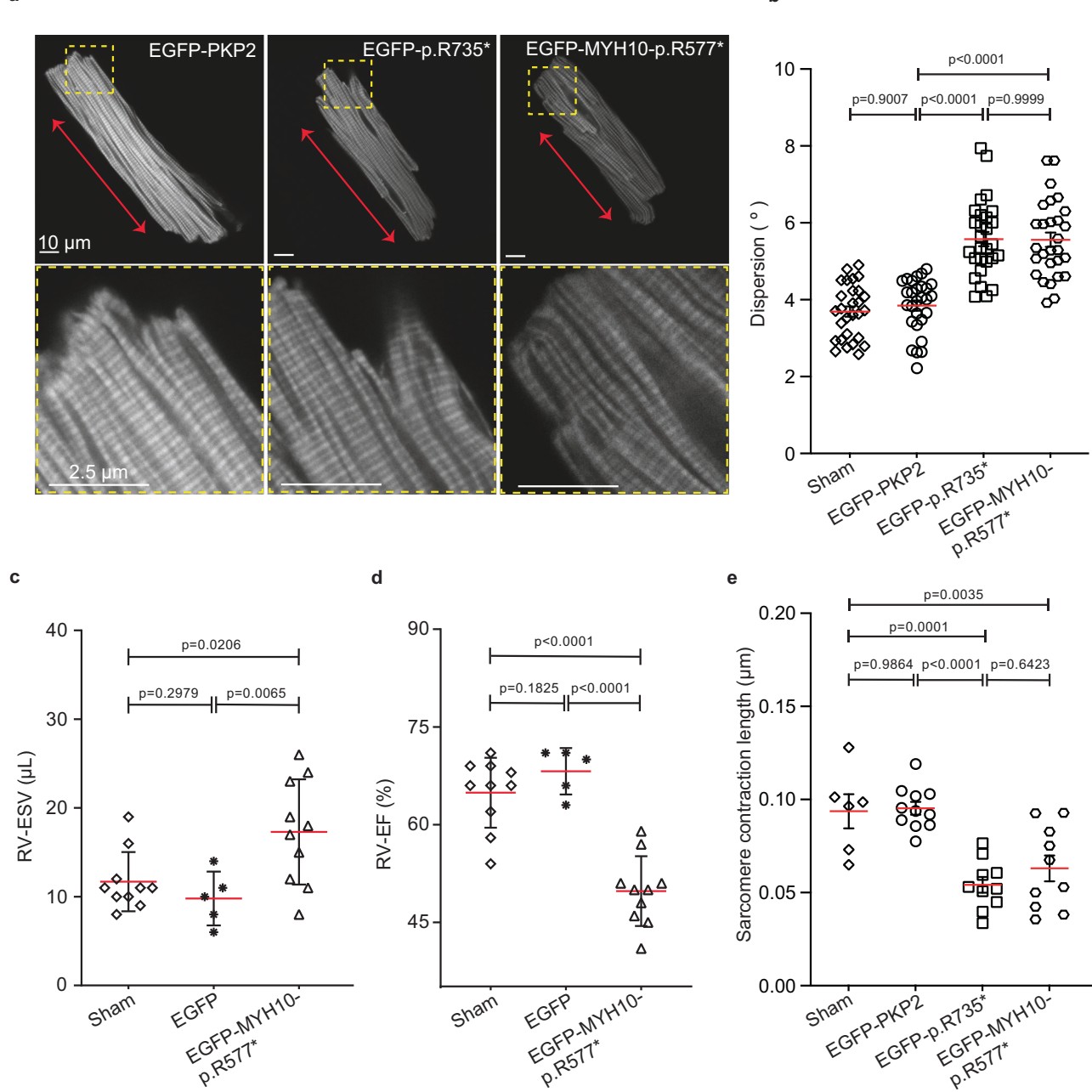

**Fig. 7 | Mice encoding PKP2-p.R735* or MYH10-p.R577* mutant present sarcomere alterations and right ventricle systolic dysfunction. a** Representative images showing cardiac actin network in phalloidin-stained isolated cardiomyocytes from in vivo transduced AAV-*EGFP-PKP2*, AAV-*EGFP-PKP2-p.R735** or AAV-*EGFP-MYH10-p.R577** mice. Red arrows indicate the normal actin fibers direction. Lower panels show a magnification from yellow boxes. Scale bars, 10 μm and 2.5 μm **b** Graph quantifies the dispersion from main direction of the actin filaments in phalloidin-stained adult cardiomyocytes isolated from mice non-transduced (sham control) and transduced with AAV-*EGFP-PKP2*, AAV-*EGFP-PKP2-p.R735** or AAV- *EGFP-MYH10-p.R577**. Data are presented as mean ± sd; n = 28 cells from two independent experiments. Significance was stablished as *p* < 0.05. (one-way ANOVA with the Tukey multiple comparison post-test). **c, d** Right ventricular (RV) end systolic volume (ESV) and ejection fraction (EF) determined by MRI in

hearts of non-transduced (sham control) and AAV-*EGFP* or AAV-*EGFP-MYH10-p.R577** transduced mice. Data are presented as mean ± sd; (n = 10 mice for all the groups but EGFP n = 5), repeated twice, **e** Graph showing the sarcomere contraction length as the difference between the maximum and minimum length in ten consecutive beats from isolated cardiomyocytes of non-transduced control, AAV-*EGFP-PKP2, AAV-EGFP-PKP2-p.R735** and AAV-*EGFP-MYH10-p.R577** mice (n = 11 cells for all the groups except for sham which was n = 6). Statistical significance was determined for **b**–**e** by one-way ANOVA with the Tukey's multiple comparison post-test; for (**c, d**) by Brown-Forsythe and Welch ANOVA tests, Significance was established as *p* < 0.05. Data are presented as mean ± sem. ° degrees, μm micrometers, μL microliters, % percentage, RV right ventricle, ESV end systolic volume, EF ejection fraction. Source data are provided as a source data file.

mouse and rat neonatal heart dissociation kit protocol (Miltenyi Biotec 130-098-373). Pipetting approximately $1.6 \times 10^5$ cells per well of glass bottom Ibidi plates covered with a layer of 1% gelatin. Cells were maintained at 37 °C with 5% of $CO_2$.

### Transfection and stable cell lines

Transient transfection with cDNAs encoding EGFP, EGFP-tagged PKP2, PKP2-p.R735*, PKP2-p.W676*, PKP2-p.Y807*, MYH10 and MYH10-p.R577* proteins were performed in 8-well cell culture dishes (Ibidi)

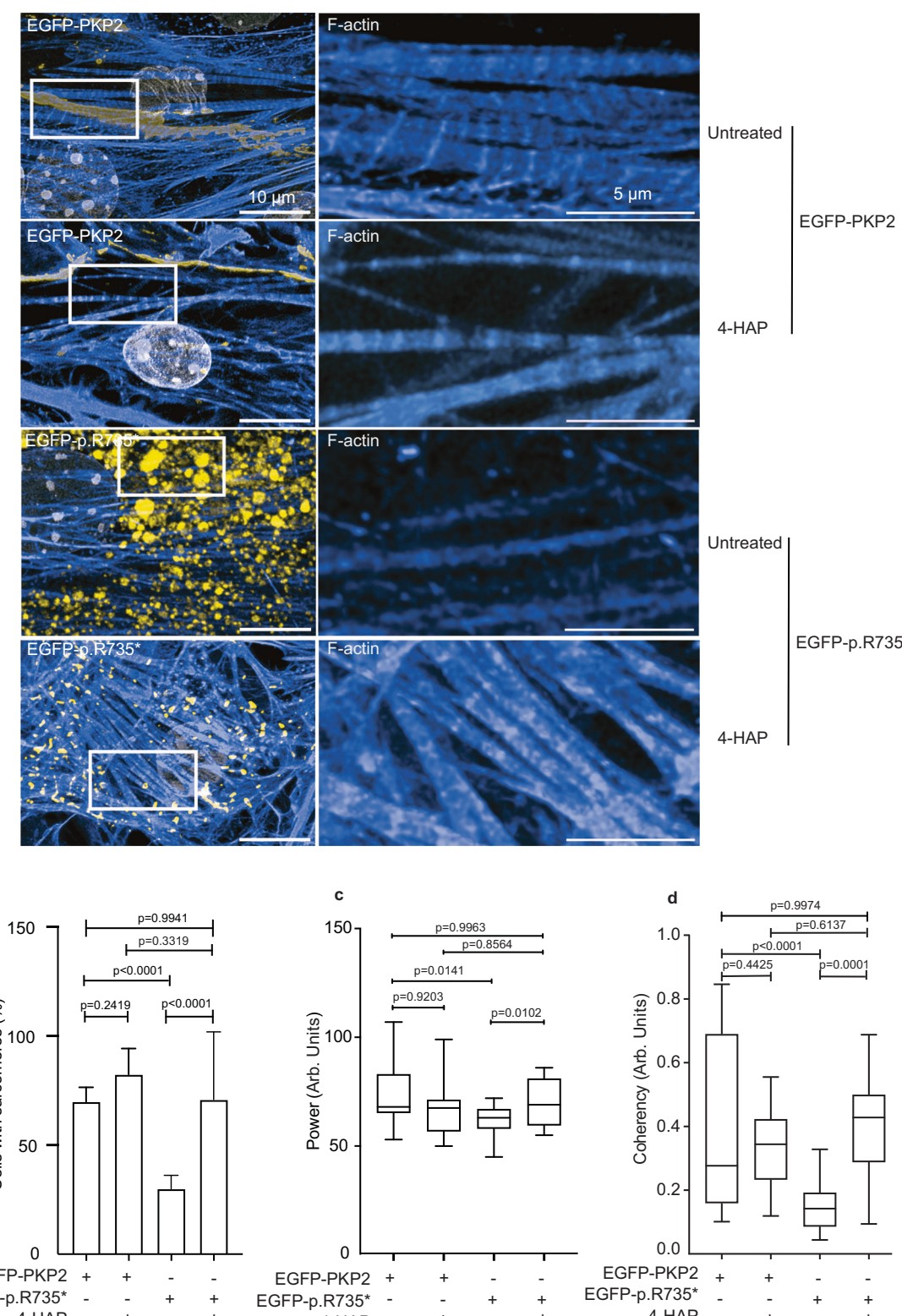

using the jetPRIME® reagent according to the manufacturer's protocol (Polyplus transfection®). Stable HL-1 cell lines were generated using the *PiggyBac* transposon system. In brief, cells were transfected with plasmids expressing *PKP2, PKP2-p.R735*, EGFP, EGFP-tagged PKP2, and PKP2-p.R735*, together with the *pPB-transposase*[50]. Cells were then selected for geneticin resistance (G418, ThermoFisher Scientific) and expanded for further experiments.

**Atomic force microscopy (AFM)**

The AFM experiments were performed with a commercial instrument (JPK NanoWizard 3, JPK Instruments AG, Germany) mounted on an Axio Vert A1 inverted microscope (Carl Zeiss, Germany). For these experiments, HL-1 cells stably expressing *PKP2* or *PKP2-p.R735* were maintained in Claycomb cell culture medium. To ensure the cells stayed alive and adherent during the force spectroscopy experiments,

**Fig. 8 | Activation of MYH10 corrects cytoskeletal defects observed in PKP2 mutant cardiomyocytes. a** Confocal representative images of F-actin (blue) in MNC expressing PKP2 or PKP2-p.R735* with or without 4-HAP treatment. Nuclei are shown in white. White boxes show a magnification of F-actin filaments organization. Scale bars, 10 μm and 5 μm. **b** Percentage of MNC with visually apparent and organized sarcomeres. Boxes depict the 25th–75th percentile with a line showing the median. Whiskers display minimum to maximum values; $n = 54$ cells for untreated EGFP-PKP2; $n = 62$ cell for EGFP-PKP2 treated with 4-HAP; $n = 79$ cells for EGFP-PKP2-p.R735*; $n = 26$ cells for EGFP-PKP2-p.R735* treated with 4-HAP examined over three independent experiments. **c** TTorg analysis of the captured images assessed the organization of the sarcomeres, and this is represented by the sarcomeric power metric. Calculated transverse organization level of sarcomeres is based on the calculation of the peak amplitude in the Fourier spectrum of the image at the sarcomeric frequency, Boxes depict the 25th–75th percentile with a line showing the median. Whiskers display minimum to maximum values; $n = 19$ ROIs for untreated EGFP-PKP2; $n = 16$ ROIs for EGFP-PKP2 treated with 4-HAP; $n = 16$ ROIs for EGFP-PKP2-p.R735* treated with 4-HAP. **d** Analysis of cellular actin coherency, using the ImageJ plugin OrientationJ. Boxes depict the 25th–75th percentile with a line showing the median. Whiskers display minimum to maximum values; $n = 28$ ROIs for untreated EGFP- PKP2; $n = 26$ ROIs for EGFP-PKP2 treated with 4-HAP; $n = 23$ ROIs for EGFP-PKP2-p.R735*; $n = 31$ ROIs for EGFP-PKP2-p.R735* treated with 4-HAP. Statistical significance was determined by two-way ANOVA with the Tukey multiple comparison post-test (**b**–**d**). Significance was established as $p < 0.05$. Data are presented as mean ± sd. μm micrometers, Arb. Units arbitrary units, % percentage. Source data are provided as a Source data file.

measurements were made at a constant temperature of 37 °C. Cell studies were conducted using specially adapted Biotool Cell XXL cantilevers (NanoandMore, Germany) with a nominal spring constant of 0.1 N/m and a height of 15 μm; the use of extra-long tips of ~15 μm minimizes accidental contact of the cell with the cantilever beam body. The half-cone angle was ~12°, and the nominal radius at the tip apex was 25 nm. The upper cantilever surface was gold-coated to improve the signal-to-noise ratio in the deflection signal. To control the force applied on the cell, the deflection sensitivity was calibrated on a Petri dish. The spring constant (0.092–1.1 N/m) was calculated using the thermal noise method[51].

Force-volume maps[24] were generated for whole cells by acquiring force-distance curves ($128 \times 128$ pixels$^2$) over a 30 μm$^2$ area. The tip sample distance was modulated by applying a triangular waveform. Each individual force-distance curve was acquired at a velocity of 100 μm/s (8 Hz) and a range of ~6 μm. To prevent sample damage, the maximum force applied to cells was 2 nN.

Maps were analyzed with in-house software written in Python. We analyzed 8 HL-1 cells expressing PKP2 and 5 cells each expressing the *PKP2-p.R735** variant from three independent experiments. The program includes bottom-effect corrections for a conical tip to correct for finite cell thickness[25].

## HaloTag pull-down and EGFP co-immunoprecipitation

PKP2 isoforms were expressed in HEK293T cells as N-terminal Halo tag fusion proteins. Cell pellets were lysed using Mammalian Lysis Buffer (G9381, Promega). The bait-prey complexes, containing the PKP2-Halo-tagged fusion protein (bait) and the potential binding partners (prey), were pulled down using HaloLink resin (Promega Madison, WI) and extensively washed in buffer containing 100 mM Tris (pH 7.6), 150 mM NaCl, 1 mg/mL BSA, and 0.05% IGEPAL® CA-630 (octylphenoxypolyethoxyethanol, I3021, Sigma-Aldrich, Oakville, ON). Purified bait-prey protein complexes were digested overnight with TEV protease at 4 °C to release Halo-linked PKP2 protein, and the tag-free protein complexes were isolated with a His-Trap-Spin column.

For EGFP co-immunoprecipitation HEK293T cells were transfected with plasmid expressing *pEGFP-PKP2* or *pEGFP-PKP2-p.R735**. After 48 hours, cells were washed with ice-cold PBS and dislodged by scraping. Proteins were extracted in NP-40 buffer (150 mM NaCl, 50 mM Tris-HCl pH 8.0, 1% NP-40, and protease and phosphatase inhibitors). Samples were incubated with shaking for 2 h at 4 °C and then centrifuged for 30 min at $20,000 \times g$ at 4 °C. All protein samples were quantified by the Lowry method (BioRad), and 1 mg of each sample were used for co-immunoprecipitation (Co-IP) and input, respectively. Co-IP was performed with Dynabeads® A (10008D, ThermoFisher Scientific). For each sample, 50 μl Dynabeads were washed four times with NP-40 buffer. After this, rat IgG1 anti-EGFP (kindly provided by the Monoclonal Antibody facility at the CNIO, Spain) were added to the Dynabeads at 1:100 dilution and incubated with shaking for 45 min at 4 °C. The antibody was then removed, and 1 mg of protein per sample was added to the Dynabeads, followed by incubation overnight with shaking at 4 °C. Unbound proteins were removed by washing the Dynabeads four times in NP-40 buffer. Bound proteins were eluted with 30 μL loading buffer (10% SDS, 10 mM β-mercapto-ethanol, 20% glycerol, 200 mM Tris-HCl pH 6.8, 0.05% Bromophenol blue), heated for 15 min at 95 °C. Proteins were western blotted with antibodies against, MYH10 (3404 S, Cell Signaling Technology, 1:1000), and EGFP (632381, Clontech, 1:1000). Secondary antibodies were HRP anti-mouse (ABIN6699027, Antibodies online, 1:4000) and anti-rabbit (ABIN5563398, Antibodies online,1:4000), as appropriate. Immunoblots were developed in the iBright 1500 system.

## Protein digestion, mass spectrometry, and peptide identification

The eluted protein complexes were in-gel digested with trypsin as described previously[52], and the resulting peptides were analyzed by liquid chromatography coupled to tandem mass spectrometry (LC-MS/MS), using an Easy nLC-1000 nano-HPLC apparatus (Thermo Scientific, San Jose, CA, USA) coupled to a hybrid quadrupole-Orbitrap mass spectrometer (Q Exactive HF, Thermo Scientific). The dried peptides were taken up in 0.1% (v/v) formic acid and then loaded onto a PepMap100 C18 LC pre-column (75 μm I.D., 2 cm, Thermo Scientific) and eluted on line onto an analytical NanoViper PepMap 100 C18 LC column (75 μm I.D., 50 cm, Thermo Scientific) with a continuous gradient consisting of 10–35% B (80% acetonitrile, 0.1% formic acid) for 60 min at 200 nL/min. Peptides were ionized using a Picotip emitter nanospray needle (New Objective). Each mass spectrometry (MS) run consisted of enhanced FT-resolution spectra (120,000 resolution) in the 400–1500 $m/z$ range followed by data-dependent MS/MS spectra of the 20 most intense parent ions acquired during the chromatographic run. For the survey scan, the AGC target value in the Orbitrap was set to 1,000,000. Fragmentation in the linear ion trap was performed at 27% normalized collision energy, with a target value of 100,000 ions. The full target was set to 30,000, with 1 microscan and 50 ms injection time, and the dynamic exclusion was set to 0.5 min. The MS/MS spectra were searched with the Sequest algorithm in Proteome Discoverer 1.4 (Thermo Scientific). The UniProt human protein database (March 2017, 158,382 entries) was searched with the following parameters: trypsin digestion with 2 maximum missed cleavage sites; precursor and fragment mass tolerances of 800 ppm and 0.02 Da, respectively; Cys carbamidomethylation as a static modification; and Met oxidation as a dynamic modification. The results were analyzed using the probability ratio method[53], and a false discovery rate (FDR) for peptide identification was calculated based on search results against a decoy database using the refined method[54].

## In-silico model of human PKP2, PKP2-p.R735*, PKP2-p.W676*, PKP2-p.Y807*, and PKP2-p.R735*-EGFP proteins

FASTA sequence of mature human PKP2 (Uniprot Id: Q99959-2) (https://www.uniprot.org/uniprotkb/Q99959/entry#Q99959-2) and its mutant forms p.W646*, p.R735*, and p.Y807* were submitted to a local implementation of I-Tasser software suite v5.1[44] for threading

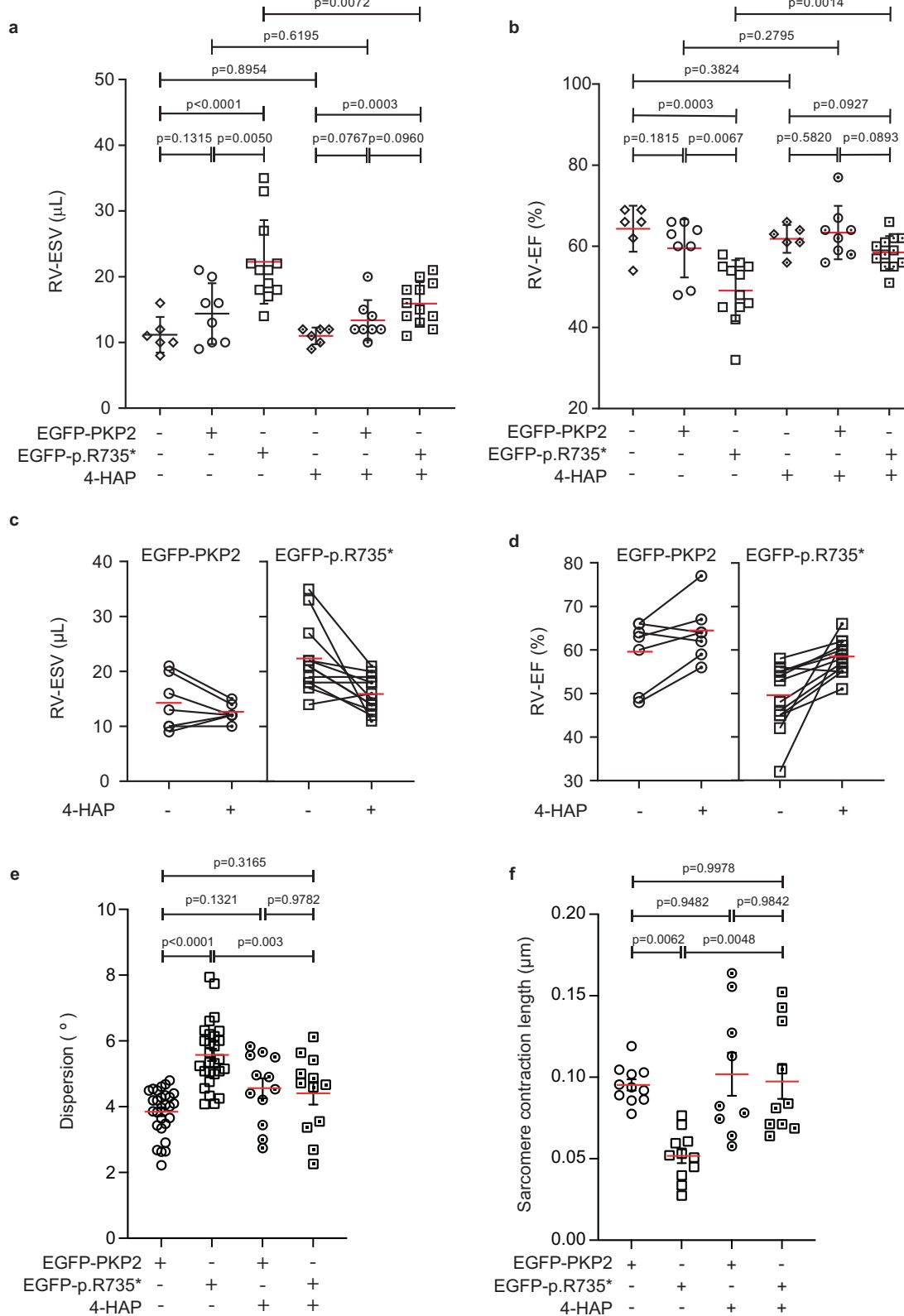

modeling. In each case, the best model with minimal energy and correct folding (best structural alignment to template pdb ID: 3tt9 corresponding to the ARM repeat; (https://www.rcsb.org/structure/3TT9) was selected. A final cycle of refinement to minimize clashes and energy was made with the relax tool[55,56] of Rosetta suite v3.8 (www.rosettacommons.org). The model with correct topology and minimal score was selected as final model.

PKP2-p.R735*-EGFP model was performed in two steps, FASTA sequence of the linker-EGFP (KPVATMVSKGEELFTGVVPILVELDGDVNG HKFSVSGEGEGDATYGKL TLKFICTTGKLPVPWPTLVTTLTYGVQCFSRYP DHMKQHDFFKSAMPEGYVQERTIFFKDDGNYKTRAEVKFEGDTLVNRIEL KGIDFKEDGNILGHKLEYNYNSHNVYIMADKQKNGIKVNFKIRHNIEDGSV QLADHYQQNTPIGDGPVLLPDNHYLSTQSALSKDPNEKRDHMVLLEFVT AAGITLGMDELYK) was submitted to a local implementation of I-Tasser

**Fig. 9 | Activation of MYH10 corrects PKP2-p.R735* contractil dysfunction.**
**a** Right ventricular (RV) end systolic volume (ESV) and **b** ejection fraction (EF) determined by MRI in hearts of non-transduced (sham) and AAV-*EGFP-PKP2* or AAV-*EGFP-PKP2-p.R735** transduced mice and treated or not with 4-HAP. Data are presented as mean ± sd; *n* = 6 mice (sham), *n* = 8 mice (EGFP-PKP2), *n* = 12 mice (EGFP-PKP2-p.R735*). Statistical significance was determined by Brown-Forsythe and Welch ANOVA tests with *p* < 0.05 considered statistically significant. **c**, **d** Graphs illustrate global RV-ESV and RV-EF in all animals analyzed before and after 4-HAP administration. Red line represents mean, and dots represent individual animals with a *n* = 7 (EGFP-PKP2) *n* = 12 (EGFP-PKP2-p.R735*). Pre-and post-4-HAP treatment values from each mouse are connected by solid lines. **e** Graph showing the dispersion from main direction of the actin filaments in phalloidin-stained

cardiomyocytes isolated from mice transduced with AAV-*EGFP-PKP2* or AAV-*PKP2-p.R735** and treated or not with 4-HAP (*n* = 12 cells from mice treated with 4-HAP and *n* = 28 cells from untreated mice). **f** Graph showing the sarcomere contraction length as the difference between the maximum and minimum length in ten consecutive beats from isolated cardiomyocytes of AAV-*EGFP-PKP2* and AAV-*EGFP-PKP2-p.R735** transduced mice, treated or not with 4-HAP. Data are presented as mean ± sem; *n* = 9 cells from mice treated with 4-HAP; *n* = 11 cells from untreated mice from three independent experiments. Statistical significance was determined by one-way ANOVA with Tukey's multiple comparison post-test with *p* < 0.05 considered statistically significant. Data are presented as mean ± sem. RV right ventricle, ESV end systolic volume, EF ejection fraction, µL microliters, % percentage, µm micrometers, ° degrees. Source data are provided as a source data file.

software suite v5.1 for threading modeling as described above[44]. The best model with minimal energy and correct folding (best structural alignment to templates previously published) was selected. For modeling the full fusion protein PKP2-p.R735*-EGFP a comparative modeling was made using de RosettaCM tool of the of Rosetta suite v3.8 (www.rosettacommons.org) with the model before, the model of PKP2-p.R735* modeled previously, the FASTA sequence of each one and the alignment with the complete sequence. The model with best structural alignment to the templates and minimal energy was selected as final candidate. A final cycle of refinement to minimize clashes and energy was made with the relax tool as before. The conformer with correct topology and minimal score was selected as final model.

### Immunostaining and imaging analysis

Mouse neonatal cardiomyocytes (MNC) and HL-1 cells were fixed with 4% formaldehyde for 5 minutes in 0.1% Triton X-100 PBS and then blocked and then blocked and permeabilized for 30 min at room temperature with 2% BSA. After that, cells were stained with Phalloidin-iFluor 594 Reagent (ab176757, Abcam, 1:1000) or Alexa Fluor™ Plus 647 Phalloidin (A30107, Invitrogen, 1:1000) for two hours. Cells were washed with phosphate buffered saline before addition of DAPI (62248, ThermoFisher Scientific, 1:1000) and were mounted in Vectashield-mounting medium). Images were acquired using the LIGHTNING module of a Leica TCS SP8 Scan head WLL confocal microscope (Leica Microsystems GmbH, Germany) equipped with an HC PL Apo CS2 63x/1.4 OIL objective. Z-stack images were captured and presented as maximal projections. Images were processed with IMARIS software (version 9.1.2, Oxford Instruments, UK). Cell height was measured from maximal projections of transverse Z-stack images. Actin and myosin filaments were analyze using ridge detection plugging in Fiji software. To estimate the local orientation of the actin fibers, we used OrientationJ (http://bigwww.epfl.ch/demo/orientation/), which is an ImageJ plugin[27]. Of every pixel of the image OrientationJ evaluates the local orientation and isotropic properties (coherency). The coherency values reflect the direction of the actin fibers in space regions of interest (ROI) of 20 × 20 µm in size; thus, values vary from 0 (completely anisotropic areas, zero directionality) to 1 (highly oriented structures). Cell images were also subjected to Fast Fourier Transformation (FFT) for analysis of sarcomere organization (ROIs size 20 × 20 µm). Sarcomeric power (the peak amplitude) in the Fourier spectrum of cardiomyocyte images was determined using ImageJ software (version 1.46) with a plugin TTorg (http://mirror.imagej.net/plugins/ttorg). The higher the sarcomeric power value, the greater the sarcomeric organization[26].

To calculate ratio membrane-cytoplasm fluorescence, z-stacks images from MNC or HL-1 cells encoding TdTomato-PKP2, EGFP-PKP2, EGFP-PKP2-p.R735*, PKP2-p.R735*-EGFP, EGFP-PKP2-p.W676* or EGFP-PKP2-p.Y807* were acquired with a Leica SP8 Navigator confocal microscope with HC PL Apo CS2 63x/1.4 OIL objective. Regions of interest (ROIs) were drawn over maxima projections images to define the plasma membrane and the cytoplasm, excluding the nucleus and/or vacuoles. The ratio plasma membrane-cytoplasm intensity was

calculated to normalize the intensity of the plasma membrane to the level of expression on each single cell.

### Heart immunodetection

Hearts were collected and fixed in 4% paraformaldehyde in phosphate-buffered saline (PBS) for overnight at 4 °C. After, these hearts were incubated in 70% ethanol overnight at 4 °C. Samples were included in paraffin and cross-sections (5 µm) were then prepared. Tissues were prepared and stained following the indications of the DAB substrate kit (Vector Laboratories). Anti GFP (R1091P, ORIGENE, 1:500) was used as the primary antibody. Images were acquired using an Olympus BX51 microscope using the fitted 10x or 20x UPlanSApo objectives and Cell Sens Entry Ink acquisition software. Images were analyzed using ImageJ and were processed for presentation with GraphPad.

Protein extraction from ventricular tissue samples of untransduced, EGFP, EGFP-PKP2, EGFP-PKP2-p.R735*, and PKP2-p.R735*-EGFP mice was performed in 125 mM NaCl, 50 mM Tris-HCl pH 8.0,1 mM EDTA, 1% SDS 1% NP-40, supplemented with protease and phosphatase inhibitors using a TissueLyser. Protein samples were quantified by the Lowry method (BioRad), separated on 8% SDS-PAGE gels, and western blotted with antibodies against, EGFP Living Colors (632381, Clontech, 1:1000) and GAPDH (sc-32233, Santa Cruz Biotechnology;1:2000). Secondary antibodies were HRP anti-mouse (ABIN6699027, Antibodies online, 1:4000). Immunoblots were developed in an iBright 1500 system.

### Cell fractionation

After 24 h, transfected cells were washed once with ice-cold PBS 1× and after dislodged by scraping using Buffer NP40 (50 mM Tris-HCl pH7,5; 150 mM NaCl; 1% Nonidet P 40 substitute). Cells were lysed for 30 min at 4 °C on a rotator and the lysates were cleared by centrifugation (17,949 × *g* for 30 min at 4 °C).

Plasma membrane proteins were extracted using the Plasma Membrane Protein Extraction Kit from Abcam. Proteins from membrane and cytoplasmic extracts were western blotted with antibodies against PKP2 (EB10841, Everest Biotech, 1:1000), N-cadherin (sc-59987, Santa Cruz Biotechnology, 1:1000), and GAPDH (sc-32233, Santa Cruz Biotechnology, 1:1000). Secondary antibodies were anti-goat (ABIN2169607, Antibodies online, 1:4000) and anti-mouse (ABIN6699027, Antibodies online, 1:4000) as appropriate. Immunoblots were developed in an iBright 1500 system.

### 4-hydroxyacetophenone (4-HAP) treatment

50 µM 4-hydroxyacetophenone was added to culture medium 48 h before performing any experiment. Animals were intraperitoneal treated with 1 mg/kg of 4-hydroxyacetophenone for seven consecutive days.

### Statistics

Statistical analysis was performed using the GraphPad Prism 9 software. All the experimental outcomes have been analyzed blindly. Mouse experiments were designed to minimize the number of animals

needed to give sufficient statistical power. No data were excluded from the analysis. When sterically possible, data are shown in dot plots to demonstrate data distribution and represent individual data points. Data are shown as mean and standard deviation (sd) or standard error of the mean (sem). Comparisons of the groups were performed by Student $t$ test, or Student $t$ test witch Welch´s correction, or unpaired two-tailed Mann–Whitney, or one-way Brown–Forsythe and Welch ANOVA tests, one-way ANOVA or two-way ANOVA with the Tukey multiple comparison post-test. Significance was established as $p < 0.05$.

### Reporting summary

Further information on research design is available in the Nature Portfolio Reporting Summary linked to this article.

## Data availability

The authors declare that the data supporting the findings of this study are available within the paper and in the Supplementary Information. Should any raw data files be needed in another format they are available from the corresponding author upon request. Source data are provided with this paper.

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

## Acknowledgements

This study was supported by MCIU grant BFU2016-75144-R and PID2020-116935RB-I00, and by a "la Caixa" Banking Foundation grant under the project code HR18-00304" to J.A.B.; The study was also supported by the "Ayudas a la Investigación Cátedra Real Madrid-Universidad Europea" (2017/RM01). C.M.-L. and S.S. hold MCIU predoctoral contracts BES-2017-079715, and BES-2017-079707 respectively. R.G. acknowledges funding from the European Research Council under grant ERC-AG-340177 (3DNanoMech) and from the MCIU under grant MAT2016-76507-R. The CNIC is supported by the Instituto de Salud Carlos III (ISCIII), the Ministerio de Ciencia e Innovación (MCIN) and the Pro CNIC Foundation and is a Severo Ochoa Center of Excellence, grant CEX2020-001041-S funded by MICIN/AEI/10.13039/501100011033. The microscopy experiments were carried out at the Dynamic Microscopy and Image Unit, CNIC, ICTS-ReDib, co-financed by MCIN/AEI /10.13039/ 501100011033 and FEDER "A way of making Europe" (#ICTS-2018-04-CNIC-16). Imaris full analysis were carried out at the Microscopy & Dynamic Imaging, CNIC, ICTS-ReDib, co-funded by MCIN/AEI /10.13039/501100011033. Biomedical Imaging has been conducted at the Advanced Imaging Unit of the CNIC (Centro Nacional de Investigaciones Cardiovasculares Carlos III), Madrid, Spain. This project used the ReDIB ICTS infrastructure TRIMA@CNIC, Ministerio de Ciencia e Innovación (MCIN).

## Author contributions

N.G.-Q., S.S., C.M.-L., and J.A.B. designed and interpreted the experimental work to characterized the PKP2 mutant models, and cells with the help of C.S.-R., A.G.-G., M.L., M.A.-S., and M.I.-G. C.M.-L., D.M.-P., D.S.-R., M.R.-M., and A.G.-G. constructed the plasmids and generated the cell lines. F.MdB. generated the in silico protein models. E.C. was responsible for mass spectrometry analyses. A.G.-G., D.M., M.L., and M.I.-G. helped with the animal work. D.S. performed the AFM experiment and analyzed the data. R.G. designed, analyzed, and interpreted the AFM experiment. J.A.B. conceived the project and wrote the manuscript. All authors discussed the results and commented on the manuscript.

## Competing interests

The authors declare no competing interests.
