## [Peer Review File · Nature Communications]

Myh10 activation rescues contractile defects in arrhythmogenic cardiomyopathy (ACM)Editorial Note: This manuscript has been previously reviewed at another journal that is not operating a transparent peer review scheme. This document only contains reviewer comments and rebuttal letters for versions considered at *Nature Communications*.

REVIEWERS' COMMENTS

Reviewer #2 (Remarks to the Author):

The authors have greatly improved the quality of the manuscript and strengthened their conclusions in this revised version. I listed many significant concerns in the previous version, but to the authors credit they have performed several new experiments, included new key controls, and performed quite a bit of quantitative image analysis that markedly strengthens key conclusions that I previously felt were insufficiently supported.

In my opinion, this revised manuscript is much improved and far more compelling, and the primary conclusions (and title) are now sufficiently supported.

One minor point - In Figure 3b, lanes 2 and 4 appear to be the same conditions, yet significantly differ from one another. I could find no clarification in the legend or text. Finally, it is another minor point (and I accept the authors claim that they can still resolve significant differences between groups with their current SNR), but I would encourage the authors to evaluate their setup and to try and improve the SNR for their adult myocyte ex vivo contraction analysis. The papers they reference as comparators are all looking at neonatal or iPS-cardiomyocytes, where the SNR will undoubtedly be lower than in adult CMs. For a more appropriate "apples to apples" comparison, please compare your supplemental figure of contractility traces with, for example, Fig 6I of Schuldt et al. *Circ HF* 2021.

Reviewers' comments:

We acknowledge reviewer #2 for taking the time to review again our manuscript and for his/her constructive criticisms and helpful suggestions.

Below we provide point-by-point responses to the reviewer's comments (in *italics*), followed by our response, and an indication of how we have addressed these comments or modified the manuscript appropriately (in blue).

Reviewer #2 (Remarks to the Author):

The authors have greatly improved the quality of the manuscript and strengthened their conclusions in this revised version. I listed many significant concerns in the previous version, but to the authors credit they have performed several new experiments, included new key controls, and performed quite a bit of quantitative image analysis that markedly strengthens key conclusions that I previously felt were insufficiently supported.

We thank Reviewer #2 for his/her very positive comments on our manuscript and the recommendations to improve it.

In my opinion, this revised manuscript is much improved and far more compelling, and the primary conclusions (and title) are now sufficiently supported.

One minor point - In Figure 3b, lanes 2 and 4 appear to be the same conditions, yet significantly differ from one another. I could find no clarification in the legend or text.

Thanks for noting this, we have now explained in the text that although the experimental setting is the same, we are measuring different fluorescent protein membrane-cytoplasm ratio; PKP2 (lanes 1 and 2) or mutant (lanes 3 and 4). We have changed the figure legend for clarity:

"b, Dot-plot showing the distribution of PKP2 versions as ratio intensity of membrane-cytoplasm. Lanes one and two show PKP2 membrane-cytoplasm ratio intensity, last two lanes show PKP2-p.R735* ratio; n=41 cells for TdTomato PKP2 transfection, n= 27 cells for TdTomato-PKP2 with EGFP-PKP2-p.R735* cotransfection; n=33 cells for EGFP-PKP2-p.R735* transfection; n=31 cells for EGFP-PKP2-p.R735* with tdTomato-PKP2 cotransfection from 3 independent experiments."

Finally, it is another minor point (and I accept the authors claim that they can still resolve significant differences between groups with their current SNR), but I would encourage the authors to evaluate their setup and to try and improve the SNR for their adult myocyte ex vivo contraction analysis. The papers they reference as comparators are all looking at neonatal or iPS-cardiomyocytes, where the SNR will undoubtedly be lower than in adult CMs. For a more appropriate "apples to apples" comparison, please compare your supplemental figure of contractility traces with, for example, Fig 6I of Schuld et al. Circ HF 2021.

We thank the reviewer for pointing out this aspect and following the reviewer's recommendation we have added this new reference for more appropriate experimental setting comparison.